# How2comm: Communication-Efficient and Collaboration-Pragmatic Multi-Agent Perception

**Dingkang Yang**[1,2†]    **Kun Yang**[1†]    **Yuzheng Wang**[1]    **Jing Liu**[1]
**Zhi Xu**[1]    **Rongbin Yin**[5]    **Peng Zhai**[1,2*]    **Lihua Zhang**[1,2,3,4*]

[1]Academy for Engineering and Technology, Fudan University
[2]Cognition and Intelligent Technology Laboratory (CIT Lab)
[3]Engineering Research Center of AI and Robotics, Ministry of Education
[4]AI and Unmanned Systems Engineering Research Center of Jilin Province
[5]FAW (Nanjing) Technology Development Company Ltd
{dkyang20, kungyang20, pzhai, lihuazhang}@fudan.edu.cn

## Abstract

Multi-agent collaborative perception has recently received widespread attention as an emerging application in driving scenarios. Despite the advancements in previous efforts, challenges remain due to various dilemmas in the perception procedure, including communication redundancy, transmission delay, and collaboration heterogeneity. To tackle these issues, we propose *How2comm*, a collaborative perception framework that seeks a trade-off between perception performance and communication bandwidth. Our novelties lie in three aspects. First, we devise a mutual information-aware communication mechanism to maximally sustain the informative features shared by collaborators. The spatial-channel filtering is adopted to perform effective feature sparsification for efficient communication. Second, we present a flow-guided delay compensation strategy to predict future characteristics from collaborators and eliminate feature misalignment due to temporal asynchrony. Ultimately, a pragmatic collaboration transformer is introduced to integrate holistic spatial semantics and temporal context clues among agents. Our framework is thoroughly evaluated on several LiDAR-based collaborative detection datasets in real-world and simulated scenarios. Comprehensive experiments demonstrate the superiority of How2comm and the effectiveness of all its vital components. The code will be released at https://github.com/ydk122024/How2comm.

## 1    Introduction

Precise perception of complex and changeable driving environments is essential to ensure the safety and reliability of intelligent agents [25, 46], *e.g.*, autonomous vehicles (AVs). With the emergence of learning-based technologies, remarkable single-agent perception systems are extensively explored for several in-vehicle tasks, such as instance segmentation [19, 58] and object detection [26, 49]. Nevertheless, single-agent perception suffers from various shortcomings due to the isolated view, such as unavoidable occlusions [55], restricted detection ranges [56], and sparse sensor observations [57]. Recently, multi-agent collaborative perception [39, 54] has provided promising solutions as an emerging application for vehicle-to-vehicle/everything (V2V/X) communications. The impressive studies [7, 14, 16, 17, 31, 32, 33, 38, 40, 51] have progressively presented to aggregate valuable information and complementary perspectives among on-road agents, resulting in a more

---

[†]Equal contributions 🐾. The two first authors thank Runsheng Xu for providing constructive suggestions.
[*]Corresponding authors.

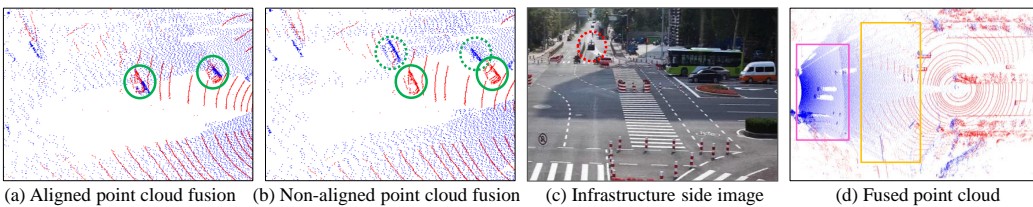

| (a) Aligned point cloud fusion | (b) Non-aligned point cloud fusion | (c) Infrastructure side image | (d) Fused point cloud |

Figure 1: (a) and (b) show the point cloud fusion results in the absence and presence of transmission delay, respectively. (c) and (d) show the vanilla image and fused point cloud of the collaborative perception scene containing an ego vehicle (red circle) and an infrastructure, respectively.

precise perception. Despite recent advancements, challenges remain due to unpreventable dilemmas, including communication redundancy, transmission delay, and collaboration heterogeneity.

**Communication Redundancy.** The dominant patterns for reducing communication overhead are summarized as feature compression [14, 33, 40] and spatial filtering [7, 8]. The former assumes that agents share all spatial areas indiscriminately, which dramatically wastes bandwidth. The latter overly relies on confidence maps to highlight gullible locations and fails to consider spatially holistic information. Moreover, these methods invariably cause losses of transmitted valuable information.

**Transmission Delay.** Figures 1(a)&(b) present the point cloud fusion results from an ego vehicle and an infrastructure in the time-synchronous and time-asynchronous cases, respectively. The inevitable transmission delay causes position misalignment of fast-moving objects within the green circles, potentially harming subsequent collaboration performance. Although several delay-aware strategies [13, 40, 53] are proposed to tackle this issue, they either suffer from performance bottlenecks [13, 40] or introduce massive computation costs [53], leading to sub-optimal solutions.

**Collaboration Heterogeneity.** Figures 1(c)&(d) show the typical collaboration scenario involving two agents and the fused point cloud. Intuitively, LiDAR configuration discrepancies (*e.g.*, different LiDAR densities, distributions, reflectivities, and noise interference) across agents potentially cause collaboration heterogeneity within the feature space. In this case, the orange box contains the common perception region of both agents, which facilitates bridging the feature-level gap caused by sensor configuration discrepancies [37, 40]. The magenta box contains the exclusive perception region of the infrastructure, which provides complementary information for the ego vehicle and compensates for the occluded view. Fusing valuable spatial semantics from these two perception regions facilitates comprehensive and pragmatic perception. However, most previous methods [14, 16, 32, 33, 40] integrate collaborator-shared features via per-agent/location message fusion to enhance ego representations, whose collaboration processes could be vulnerable since the advantages of distinct perception regions from heterogeneous agents are not considered holistically. Moreover, the current single-frame perception paradigm faces the challenges of 3D point cloud sparsity and localization errors, increasing the difficulty of building a robust multi-agent perception system.

Motivated by the above observations, we propose *How2comm*, an end-to-end collaborative perception framework to address the existing issues jointly. Through three novel components, How2comm advances towards a reasonable trade-off between perception performance and communication bandwidth. Specifically, **(i)** we first design a mutual information-aware communication mechanism to maximally preserve the beneficial semantics from vanilla characteristics in the transmitted messages of collaborators. In this case, spatial-channel message filtering is introduced to determine *how* to use less bandwidth for efficient communication. **(ii)** Second, we present a flow-guided delay compensation strategy to predict the future features of collaborators by mining contextual dependencies in sequential frames. Our ingenious strategy determines *how* to dynamically compensate for the delay's impact and explicitly accomplish temporal alignment. **(iii)** Furthermore, we construct a spatio-temporal collaboration transformer (STCFormer) module to integrate perceptually comprehensive information from collaborators and temporally valuable clues among agents. Our unified transformer structure determines *how* to achieve pragmatic collaboration, contributing to a more robust collaborative perception against localization errors and feature discrepancies. How2comm is systematically evaluated on various collaborative 3D object detection datasets, including DAIR-V2X [52], V2XSet [40], and OPV2V [41]. Quantitative experiments demonstrate that our framework significantly outperforms previous state-of-the-art (SOTA) methods under the bandwidth-limited noisy setting. Systematic analyses confirm the robustness of How2comm against distinct collaboration noises.

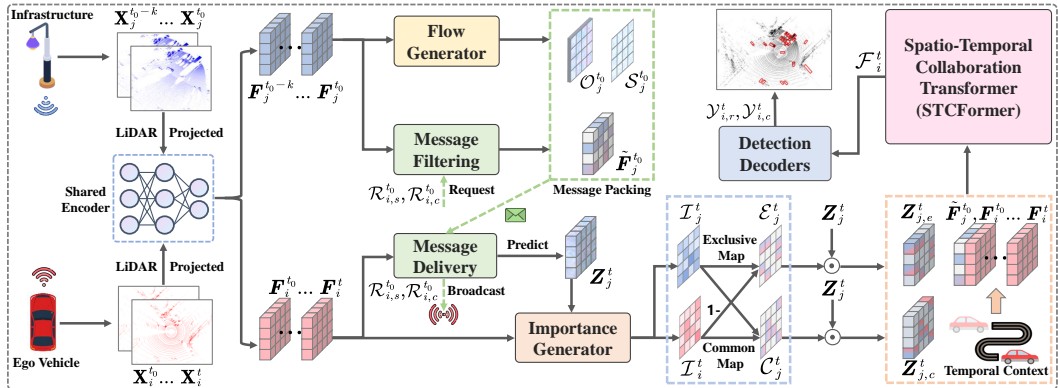

Figure 2: How2comm overview. The projected features of all agents are obtained via a shared encoder. Upon receiving requests $\{\mathcal{R}_{i,s}^{t_0}, \mathcal{R}_{i,c}^{t_0}\}$ from the ego vehicle, the collaborator (*i.e.*, infrastructure) shares the sparse feature $\tilde{\boldsymbol{F}}_j^{t_0}$, feature flow $\mathcal{O}_j^{t_0}$, and scale matrix $\mathcal{S}_j^{t_0}$ via the message filtering and flow generator. After that, the ego vehicle predicts the future feature $\boldsymbol{Z}_j^t$ and adopts the importance maps $\{\mathcal{I}_i^t, \mathcal{I}_j^t\}$ to get the exclusive and common maps, which decouple $\boldsymbol{Z}_j^t$ into $\boldsymbol{Z}_{j,e}^t$ and $\boldsymbol{Z}_{j,c}^t$. Finally, our STCFormer fuses the temporal context and decoupled spatial features to output $\mathcal{F}_i^t$ used for detection.

## 2 Related Work

**Multi-Agent Communication.** Benefiting from rapid advances in learning-based technologies [12, 21, 22, 23, 24, 42, 43, 44, 45, 47, 48, 50], communication has played an essential role in constructing robust and stable multi-agent systems. Although early works provided heuristic insights into information sharing among different agents through predefined protocols and centralized paradigms [4, 28, 29], these efforts are typically difficult to generalize into challenging scenarios. Recently, several learning-driven communication strategies have been proposed to accommodate diverse scenario applications. For instance, Vain [6] utilized an attentional neural structure to specify what information needs to be shared in agent interactions. ATOC [9] introduced the recurrent unit to decide whom the agent communicates with by receiving local observations and action intentions from other agents. TarMAC [2] designed a reinforcement learning-oriented architecture to learn communication from task-specific rewards. In comparison, we focus on LiDAR-based collaborative 3D object detection tasks. For more challenging driving scenarios, we design the mutual information supervision and attention-guiding mechanism to achieve efficient communication across agents.

**Collaborative Perception.** Collaborative perception is only in its infancy as a promising application of multi-agent systems. Several impressive approaches have been designed to facilitate the overall perception performance of AVs. The mainstream works [7, 14, 16, 33, 37, 38, 40] followed the intermediate collaboration pattern to balance average precision and bandwidth overhead. Specifically, When2com [16] introduced a handshake mechanism to determine when to communicate with collaborators. V2VNet [33] employed a fully connected graph network to aggregate feature representations shared by agents. After that, DiscoNet [14] proposed a knowledge distillation framework to supervise the intermediate collaboration through an early collaboration of full views. V2X-ViT [40] designed distinct attention interactions to facilitate adaptive information fusion among heterogeneous agents. Where2comm [7] aimed to transmit perceptually critical information via sparse spatial confidence maps. However, these methods invariably ignore valuable historical clues and lead to sub-optimal solutions. In this paper, we propose a novel collaboration transformer to jointly capture spatial semantics and temporal dynamics among agents, resulting in a more pragmatic collaboration.

## 3 Methodology

### 3.1 Problem Formulation

In this paper, we seek to develop a communication-efficient and collaboration-pragmatic multi-agent system to enhance the perception ability of the ego agent. Figure 2 illustrates the proposed system framework, which accommodates different agents (*e.g.*, AVs and infrastructures). Consider $N$ agents in a driving scene, let $\boldsymbol{X}_i^t$ be the local point cloud observation of the $i$-th agent (*i.e.*, ego agent) and

$\mathcal{Y}_i$ be the corresponding ground-truth supervision. The objective of How2comm is to maximize the LiDAR-based 3D detection performance $\Re(\cdot)$ under a total communication budget $B$:

$$\Re(B) = \underset{\theta, \tilde{\boldsymbol{F}}_j}{\arg\max} \sum_i^N \eth(\Psi_\theta(\boldsymbol{X}_i^t, \{\tilde{\boldsymbol{F}}_j^{t_0}\}_{j=1}^N), \mathcal{Y}_i), \quad \text{s.t.} \sum_j |\tilde{\boldsymbol{F}}_j^{t_0}| \le B, \quad (1)$$

where $\eth(\cdot, \cdot)$ denotes the perception evaluation metric and $\Psi_\theta$ is the perception system parameterized by $\theta$. $\tilde{\boldsymbol{F}}_j^{t_0}$ is the message transmitted from the $j$-th agent to the $i$-th agent at time delay $\tau$-aware moment $t_0$, where $t_0 = t - \tau$. The remainder of Section 3 details the major components.

## 3.2 Metadata Conversion and Feature Extraction

In the initial stage of collaboration, we build a communication graph [14, 40] where one agent is selected as the ego agent and the other connected agents act as collaborators. Upon receiving the broadcast metadata (*e.g.*, poses and extrinsic) from the ego agent, the collaborators project their local observations to the ego agent's coordinate system. Moreover, ego-motion compensation [35] synchronizes each agent's historical frames. The shared PointPillar [11] encoder $f_{enc}(\cdot)$ converts the point cloud of the $i$-th agent at timestamp $t$ into the bird's-eye-view (BEV) features as $\boldsymbol{F}_i^t = f_{enc}(\boldsymbol{X}_i^t) \in \mathbb{R}^{H \times W \times C}$, where $H, W, C$ denote height, width, and channel, respectively.

## 3.3 Mutual Information-aware Communication

Previous attempts to reduce the required transmission bandwidth relied heavily on autoencoders [14, 33, 40] or confidence maps [7, 8], which are one-sided as they only consider information compression over spatial positions or channels. To this end, we design a mutual information-aware communication (MIC) mechanism to select the most informative messages from space and channels to save precious bandwidth. MIC consists of two core parts as follows.

**Spatial-channel Message Filtering.** Stemming from solid evidence in signal picking, we first introduce the CBAM [34]-like spatial-channel attention queries to assist each agent in sharing their salient features. The spatial query $\mathcal{A}_{i,s}^{t_0} = \sigma(\omega_{3*3}[\varphi_a(\boldsymbol{F}_i^{t_0}); \varphi_m(\boldsymbol{F}_i^{t_0})]) \in \mathbb{R}^{H \times W \times 1}$ reflects what spatial locations on delayed feature $\boldsymbol{F}_i^{t_0}$ are informative, where $[\cdot ; \cdot]$ is the concatenation, $\sigma$ is the sigmoid activation, $\varphi_{a/m}(\cdot)$ denote average and max pooling functions, and $\omega_{3*3}(\cdot)$ is the 2D $3 \times 3$ convolution operation. The channel query $\mathcal{A}_{i,c}^{t_0} = \sigma(\omega_{1*1}(\varphi_a(\boldsymbol{F}_i^{t_0})) + \omega_{1*1}(\varphi_m(\boldsymbol{F}_i^{t_0}))) \in \mathbb{R}^{1 \times 1 \times C}$ reflects which channels in $\boldsymbol{F}_i^{t_0}$ are semantically meaningful. $\varphi_{a/m}(\cdot)$ in the spatial and channel queries are applied to the channel and spatial dimensions, respectively. Then, the ego agent indicates the supplementary messages required to improve local perception performance by broadcasting request queries $\mathcal{R}_{i,s}^{t_0}/\mathcal{R}_{i,c}^{t_0} = 1 - \mathcal{A}_{i,s}^{t_0}/\mathcal{A}_{i,c}^{t_0}$. The $j$-th collaborator then aggregates the requests with its attention queries to obtain a spatial-channel binary message filtering matrix as follows:

$$\mathcal{M}_j^{t_0} = f_{sel}(\omega_{1*1}[\mathcal{R}_{i,s}^{t_0}; \mathcal{A}_{j,s}^{t_0}] \odot \omega_{1*1}[\mathcal{R}_{i,c}^{t_0}; \mathcal{A}_{j,c}^{t_0}]) \in \{0, 1\}^{H \times W \times C}, \quad (2)$$

where $f_{sel}(\cdot)$ is a threshold-based selection function and $\odot$ is the element-wise multiplication. Ultimately, the selected feature map is obtained as $\tilde{\boldsymbol{F}}_j^{t_0} = \boldsymbol{F}_j^{t_0} \odot \mathcal{M}_j^{t_0}$, which provides spatial-channel sparse, yet perceptually critical information.

**Mutual Information Maximization Supervision.** Most existing works ignore the potential loss of valuable information due to feature compression. To overcome this dilemma, we maximally sustain the local critical semantics in the corresponding vanilla feature $\boldsymbol{F}_j^{t_0}$ on the selected regions of the transmitted features $\tilde{\boldsymbol{F}}_j^{t_0}$ by mutual information estimation. Since we only focus on maximizing the mutual information rather than getting its precise value, a stable estimator [5] is utilized to build the objective supervision based on the Jensen-Shannon divergence. Formally, the mutual information between two random variables $\mathcal{X}$ and $\mathcal{Z}$ is estimated as follows:

$$\hat{\boldsymbol{I}}_\varrho^{(JSD)}(\mathcal{X}, \mathcal{Z}) = \mathbb{E}_{p(x,z)}[-log(1 + e^{-T_\varrho(x,z)})] - \mathbb{E}_{p(x)p(z)}[log(1 + e^{T_\varrho(x,z)})], \quad (3)$$

where $T_\varrho : \mathcal{X} \times \mathcal{Z} \to \mathbb{R}$ is a statistics network parameterized by $\varrho$. In our case, the mutual information supervision of all collaborators within the communication link is defined as follows:

$$\mathcal{L}_{mul} = \frac{1}{N-1} \sum_{j \in \{1,...,N\}, j \ne i} \hat{\boldsymbol{I}}_\varrho^{(JSD)}(\boldsymbol{F}_j^{t_0}, \tilde{\boldsymbol{F}}_j^{t_0}). \quad (4)$$

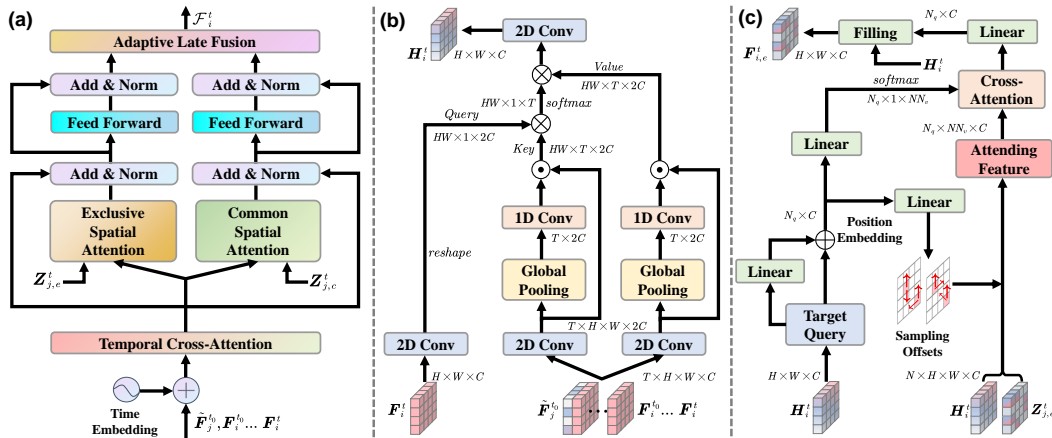

Figure 3: (a) The overall architecture of the proposed STCFormer. (b) and (c) show the structure of the temporal cross-attention (TCA) and exclusive spatial attention (ESA) modules, respectively, which contain the computational flow and the major dimensional transformations of features.

### 3.4 Flow-guided Delay Compensation

We present a flow-guided delay compensation (FDC) strategy to eliminate the two-sided fusion error in feature-level collaboration due to temporal asynchrony. Existing solutions relied on received historical features [13] and produced large errors under severe delay [53], leading to performance bottlenecks. To tackle the issues, we adopt the philosophy of feature flow to predict the collaborators' future features for temporal alignment with the ego representation. The details are as follows.

**Flow Generation and Warping.** Due to the uncertain delay between the ego agent and collaborators, FDC predicts the feature flow $\mathcal{O}_j^{t_0}$ for a fixed time interval and scale matrix $\mathcal{S}_j^{t_0}$ based on the $j$-th agent's historical frames. Specifically, as Figure 2 shows, features $\{\boldsymbol{F}_j^{t_0-k}, ..., \boldsymbol{F}_j^{t_0}\}$ are concatenated in channel dimension and entered to a generator $f_{flow}(\cdot)$ to output $\mathcal{O}_j^{t_0} \in \mathbb{R}^{H \times W \times 2}$ and $\mathcal{S}_j^{t_0} \in \mathbb{R}^{H \times W \times 1}$. Then the $j$-th agent sends $\{\mathcal{O}_j^{t_0}, \mathcal{S}_j^{t_0}\}$ with prediction ability and sparse feature $\tilde{\boldsymbol{F}}_j^{t_0}$ to the ego agent. The ego agent estimates predicted collaborator features as $\boldsymbol{Z}_j^t = f_{warp}(\tilde{\boldsymbol{F}}_j^{t_0}, (t - t_0) \cdot \mathcal{O}_j^{t_0}) \odot \mathcal{S}_j^{t_0}$, where $f_{warp}(\cdot)$ is the bilinear warping function applied to all positions and channels [60, 61], and $\cdot$ is the scalar multiplication. The temporally aligned features are passed to the STCFormer.

**Self-supervised Training Pattern.** Self-supervised learning is employed to train the flow generator $f_{flow}(\cdot)$ since the existing datasets [40, 41, 52] lack the motion annotations. Concretely, we first form the training group $\{\boldsymbol{F}_j^{t_0-k}, ..., \boldsymbol{F}_j^{t_0}, \boldsymbol{F}_j^t\}$, where $\{\boldsymbol{F}_j^{t_0-k}, ..., \boldsymbol{F}_j^{t_0}\}$ is a continuous feature sequence, and $\boldsymbol{F}_j^t$ is considered as the ground truth feature. Subsequently, we predict the feature $\boldsymbol{Z}_j^t$ as $\boldsymbol{Z}_j^t = f_{warp}(\boldsymbol{F}_j^{t_0}, (t - t_0) \cdot \mathcal{O}_j^{t_0}) \odot \mathcal{S}_j^{t_0}$. Since the optimization objective of $f_{flow}(\cdot)$ is to increase the similarity between $\boldsymbol{F}_j^t$ and $\boldsymbol{Z}_j^t$, we formulate the self-supervised loss function $\mathcal{L}_{flow}$ based on the cosine similarity [53] as follows:

$$\mathcal{L}_{flow} = \frac{1}{N-1} \sum_{j \in \{1,...,N\}, j \neq i} \left(1 - \frac{\boldsymbol{F}_j^t \odot \boldsymbol{Z}_j^t}{\|\boldsymbol{F}_j^t\|_F^2 \cdot \|\boldsymbol{Z}_j^t\|_F^2}\right), \tag{5}$$

where $\| \cdot \|_F^2$ is the squared Frobenius norm.

### 3.5 Spatio-Temporal Collaboration Transformer

To efficiently mitigate collaboration heterogeneity, we propose a spatio-temporal collaboration transformer (STCFormer) to jointly integrate the decoupled spatial semantics and temporal dynamics among agents. From Figure 3(a), the core contributions of STCFormer lie in the following three customized modules, where the other basic components follow the choice of the vanilla transformer [30].

**Temporal Cross-Attention.** To bridge the detection gap regarding fast-moving objects due to point cloud sparsity, we capture historical context clues across agents to reinforce the current

representation via a temporal cross-attention (TCA) module. The core is to perform Query-Key-Value-like attention operations by projecting the ego feature $\boldsymbol{F}_i^t$ and merged temporal features $\boldsymbol{E} = [\tilde{\boldsymbol{F}}_j^{t_0}, \boldsymbol{F}_i^{t_0}, ..., \boldsymbol{F}_i^t]$ into different subspaces via three 2D convolutional layers $\omega_{3*3}(\cdot)$. In Figure 3(b), the branches of Key&Value $\boldsymbol{E}_{k/v} \leftarrow \omega_1(\varphi_a(\omega_{3*3}(\boldsymbol{E}))) \odot \omega_{3*3}(\boldsymbol{E})$ share the same structure but separate weights, where a 1D temporal convolution $\omega_1(\cdot)$ with global average pooling $\varphi_a(\cdot)$ provides temporal correlations. $\varphi_a(\cdot)$ is applied to the spatial dimension to shrink the feature map. The computation of TSA can be shown as:

$$\boldsymbol{H}_i^t = \omega_{3*3}(\mathrm{softmax}(\omega_{3*3}(\boldsymbol{F}_i^t) \otimes \boldsymbol{E}_k^T) \otimes \boldsymbol{E}_v) \in \mathbb{R}^{H \times W \times C}. \tag{6}$$

**Decoupled Spatial Attention.** To comprehensively integrate the distinct spatial semantics from the collaborators, we facilitate pragmatic message fusion with a feature decoupling perspective inspired by the observation in Figure 1(d). Formally, an importance generator $f_{gen}(\cdot)$ is employed to generate the importance maps of the ego feature $\boldsymbol{F}_i^t$ and the estimated collaborator feature $\boldsymbol{Z}_j^t$ as $\mathcal{I}_i^t/\mathcal{I}_j^t = \sigma(\varphi_m(f_{gen}(\boldsymbol{F}_i^t/\boldsymbol{Z}_j^t))) \in [0,1]^{H \times W}$. The importance maps reflect the perceptually critical level of each pixel in the features. Then, the $j$-th agent spatially decouples the feature $\boldsymbol{Z}_j^t$ via candidate maps $\mathcal{E}_j^t = (1 - \mathcal{I}_i^t) \odot \mathcal{I}_j^t$ and $\mathcal{C}_j^t = \mathcal{I}_i^t \odot \mathcal{I}_j^t$. Intuitively, $\mathcal{E}_j^t$ and $\mathcal{C}_j^t$ depict the collaborators' exclusive and common perception regions relative to the ego agent, respectively. The exclusive and common collaborator features are obtained as $\boldsymbol{Z}_{j,e}^t = f_{sel}(\mathcal{E}_j^t) \odot \boldsymbol{Z}_j^t$ and $\boldsymbol{Z}_{j,c}^t = f_{sel}(\mathcal{C}_j^t) \odot \boldsymbol{Z}_j^t$.

Then, we present two spatial attention modules based on deformable cross-attention [59] to aggregate the decoupled exclusive and common features, which share the same structure but different weights. Here the exclusive spatial attention (ESA) is taken as an example (see Figure 3(c)), and its input comprises $\boldsymbol{H}_i^t$ and $\boldsymbol{Z}_{j,e}^t$. An importance-aware query initialization is first designed to guide ESA to focus on the potential foreground objects. Specifically, we obtain the element-wise summation of the importance maps as $\mathcal{I}^t = \sum_{j=1}^N \mathcal{I}_j^t$ and extract $N_q$ target queries from the salient locations in $\mathcal{I}^t$. The attention scores are learned from the initial queries via a linear layer and the softmax function. Subsequently, a linear layer learns an offset map for each input feature, providing the 2D spatial offset $\{\Delta q_v \,|\, 1 \le v \le N_v\}$ for each query $q$. We sample the keypoints based on the learned offset maps and extract these keypoints' features to form the attending feature. The cross-attention layer aggregates multiple collaborators' features to output the enhanced feature for each query $q$ as:

$$ESA(q) = \sum_{u=1}^U \mathcal{W}_u [\sum_{j=1}^N \sum_{v=1}^{N_v} \mathrm{softmax}(\mathcal{W}_f \boldsymbol{H}_i^t(q)) \boldsymbol{Z}_{j,e}^t(q + \Delta q_v)], \tag{7}$$

where $u$ indexes the attention head, and $\mathcal{W}_{u/f}$ denotes the learnable parameters. Then, the filling operation fills $ESA(q)$ into $\boldsymbol{H}_i^t$ based on the initial positions of the queries and outputs $\boldsymbol{F}_{i,e}^t$. Similarly, the enhanced common feature $\boldsymbol{F}_{i,c}^t$ is obtained via the common spatial attention (CSA).

**Adaptive Late Fusion.** The adaptive late fusion (ALF) module is presented to effectively fuse the exclusive and common representations $\{\boldsymbol{F}_{i,e}^t, \boldsymbol{F}_{i,c}^t\}$ for incorporating their perceptual advantages. Formally, we obtain two weight maps as $\mathcal{G}_{i,e}^t/\mathcal{G}_{i,c}^t = \omega_{1*1}(\boldsymbol{F}_{i,e}^t/\boldsymbol{F}_{i,c}^t)$, and apply the softmax function to produce the normalized weight maps as $\boldsymbol{G}_{i,e}^t/\boldsymbol{G}_{i,c}^t = \mathrm{softmax}(\mathcal{G}_{i,e}^t/\mathcal{G}_{i,c}^t)$. The learned $\boldsymbol{G}_{i,e}^t$ and $\boldsymbol{G}_{i,c}^t$ reflect the complementary perception contributions of $\{\boldsymbol{F}_{i,e}^t, \boldsymbol{F}_{i,c}^t\}$ at each spatial location. Therefore, we adaptively activate the perceptually critical information of each representation by a weighted summation. The refined feature map is obtained as $\mathcal{F}_i^t = \boldsymbol{G}_{i,e}^t \odot \boldsymbol{F}_{i,e}^t + \boldsymbol{G}_{i,c}^t \odot \boldsymbol{F}_{i,c}^t$.

### 3.6 Detection Decoders and Objective Optimization

Two detection decoders $\{f_{dec}^r(\cdot), f_{dec}^c(\cdot)\}$ are employed to convert the output fused representation $\mathcal{F}_i^t$ into the prediction results. The regression result represents the position, size, and yaw angle of the predefined box at each location, which is $\mathcal{Y}_{i,r}^{(t)} = f_{dec}^r(\mathcal{F}_i^t) \in \mathbb{R}^{H \times W \times 7}$. The classification result is $\mathcal{Y}_{i,c}^{(t)} = f_{dec}^c(\mathcal{F}_i^t) \in \mathbb{R}^{H \times W \times 2}$, revealing the confidence value of each bounding box to be an object. For objective optimization, we leverage the smooth absolute error loss for regression (denoted as $\mathcal{L}_{reg}$) and the focal loss [15] for classification (denoted as $\mathcal{L}_{cla}$). In total, we formulate the overall objective function as follows: $\mathcal{L}_{all} = \mathcal{L}_{reg} + \mathcal{L}_{cla} + \mathcal{L}_{mul} + \mathcal{L}_{flow}$.

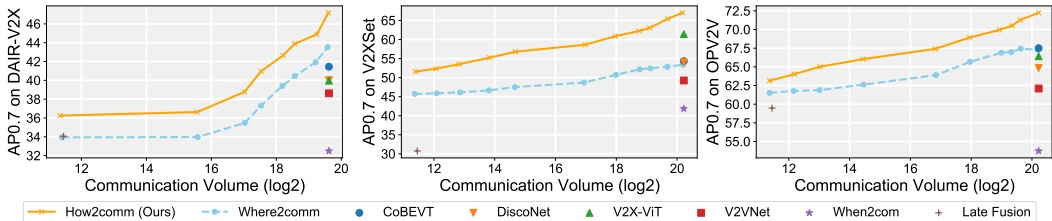

Figure 4: Collaborative perception performance comparison of How2comm and Where2comm [7] on the DAIR-V2X, V2XSet, and OPV2V datasets with varying communication volumes.

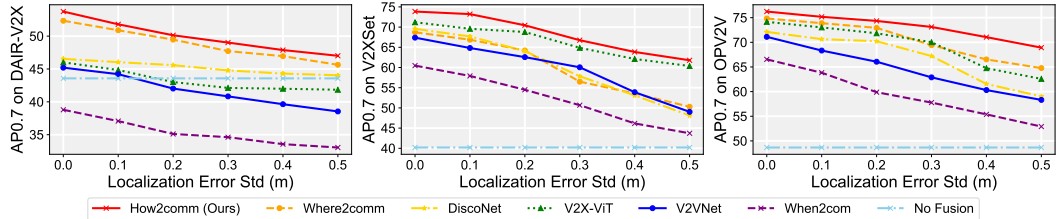

Figure 5: Robustness to the localization error on the DAIR-V2X, V2XSet, and OPV2V datasets.

# 4 Experiments

## 4.1 Datasets and Implementation Details

**Multi-Agent 3D Detection Datasets.** To evaluate the performance of How2comm on the collaborative perception task, we conduct extensive experiments on three multi-agent datasets, including DAIR-V2X [52], V2XSet [40], and OPV2V [41]. **DAIR-V2X** [52] is a real-world vehicle-to-infrastructure perception dataset containing 100 realistic scenarios and 18,000 data samples. Each sample collects the labeled LiDAR point clouds of a vehicle and an infrastructure. The training/validation/testing sets are split in a ratio of 5:2:3. **V2XSet** [40] is a simulated dataset supporting V2X perception, co-simulated by Carla [3] and OpenCDA [36]. It includes 73 representative scenes with 2 to 5 connected agents and 11,447 3D annotated LiDAR point cloud frames. The training/validation/testing sets are 6,694, 1,920, and 2,833 frames, respectively. **OPV2V** [41] is a large-scale simulated dataset for multi-agent V2V perception, comprising 10,914 LiDAR point cloud frames with 3D annotation. The training/validation/testing splits include 6,764, 1,981, and 2,169 frames, respectively.

**Evaluation Metrics.** We adopt the Average Precision (AP) at Intersection-over-Union (IoU) thresholds of 0.5 and 0.7 to evaluate the 3D object detection performance. Also, the calculation format of communication volume in [7] is used to count the message size by byte in the log scale with base 2.

## 4.2 Quantitative Evaluation

**Experimental Settings.** We build all the models using the Pytorch toolbox [18] and train them on Tesla V100 GPUs with the Adam optimizer [10]. The learning rate is set to 2e-3 and decays exponentially by 0.1 every 15 epochs. The training settings on the DAIR-V2X [52], V2XSet [40], and OPV2V [41] datasets include: the training epochs are $\{30, 40, 40\}$, and batch sizes are $\{2, 1, 1\}$. The height and width resolution of the feature encoder $f_{enc}(\cdot)$ is 0.4 m. The selection function $f_{sel}(\cdot)$ has a threshold of 0.01. The flow generator $f_{flow}(\cdot)$ leverages the multi-scale backbone to extract multi-grained representations and an extra encoder to produce $\mathcal{O}_j^{t_0}$ and $\mathcal{S}_j^{t_0}$. We implement the importance gen-

Table 1: Performance comparison on the DAIR-V2X [52], V2XSet [40], and OPV2V [41] datasets. The results are reported in AP@0.5/0.7.

| Models | DAIR-V2X AP@0.5/0.7 | V2XSet AP@0.5/0.7 | OPV2V AP@0.5/0.7 |
|---|---|---|---|
| No Fusion | 50.03/43.57 | 60.60/40.20 | 68.71/48.66 |
| Late Fusion | 48.93/34.06 | 54.92/30.75 | 79.62/59.48 |
| When2com [16] | 46.64/32.49 | 65.06/41.87 | 70.64/53.73 |
| F-Cooper [1] | 49.77/35.21 | 71.48/46.92 | 75.27/63.05 |
| AttFuse [41] | 50.86/38.30 | 70.85/48.66 | 79.14/64.52 |
| V2VNet [33] | 52.18/38.62 | 79.09/49.25 | 77.45/62.10 |
| DiscoNet [14] | 51.44/40.01 | 79.83/54.06 | 81.08/64.85 |
| V2X-ViT [40] | 51.68/39.97 | 83.64/61.41 | 80.61/66.42 |
| CoBEVT [38] | 56.08/41.45 | 81.07/54.33 | 81.59/67.50 |
| Where2comm [7] | 59.34/43.53 | 82.02/53.38 | 82.75/67.29 |
| **How2comm (ours)** | **62.36/47.18** | **84.05/67.01** | **85.42/72.24** |

erator $f_{gen}(\cdot)$ and statistical network $T_\varrho(\cdot)$ with the classification decoder in [11] and following [27], respectively. The keypoint number $N_v$ is 9, and the attention head is 8. Two $1 \times 1$ convolutional layers

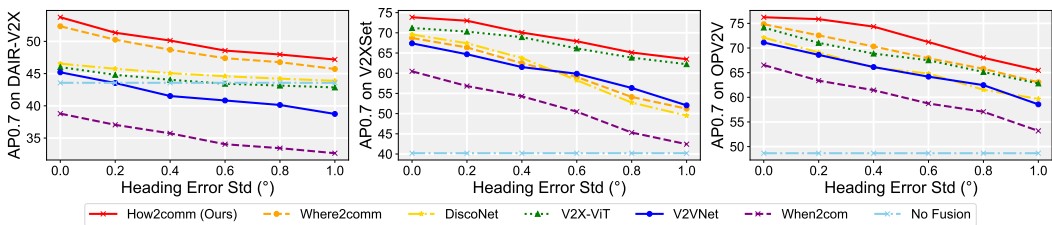

Figure 6: Robustness to the heading error on the DAIR-V2X, V2XSet, and OPV2V datasets.

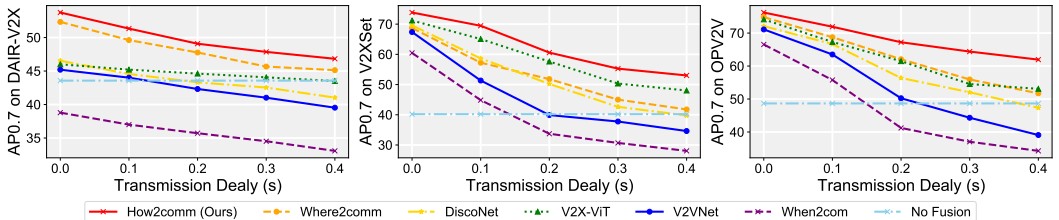

Figure 7: Robustness to the transmission delay on the DAIR-V2X, V2XSet, and OPV2V datasets.

are utilized to implement the detection decoders. Under the default noise settings, the transmission delay $\tau$ is set to 100 ms, and the localization and heading errors of the collaborators are sampled from a Gaussian distribution with standard deviations of 0.2 m and 0.2°, respectively. All experiments are constrained to $\approx$ 1 MB bandwidth consumption to reflect the narrow communication channels in real V2V/X scenarios [20].

**Detection Performance Comparison.** Table 1 compares the detection performance of the proposed How2comm with various models on three datasets under default noise settings. We consider two typical baselines. No Fusion is a single-agent perception pattern that only uses the local observations. Late Fusion integrates predicted boxes across agents and produces results with non-maximum suppression. Moreover, the existing SOTAs are comprehensively considered, including When2com [16], F-Cooper [1], AttFuse [41], V2VNet [33], DiscoNet [14], V2X-ViT [40], CoBEVT [38], and Where2comm [7]. Intuitively, How2comm outperforms previous methods in the real-world (DAIR-V2X [52]) and simulated datasets, demonstrating the superiority of our model and its robustness to various realistic noises. In particular, the SOTA performance of AP@0.7 on the DAIR-V2X and OPV2V is improved by 8.4% and 7.0%, respectively. Compared to previous per-agent/location message fusion efforts [14, 16, 33, 40], How2comm simultaneously considers the decoupled spatial semantics and temporal dynamics among agents, resulting in a more precise perception.

**Comparison of Communication Volume.** Figure 4 presents the performance comparison results with distinct bandwidth consumptions. Concretely, the orange and blue curves denote the detection precision of our How2comm and Where2comm under varying communication volumes, respectively. **(i)** How2comm keeps superior to Where2comm across all the communication choices, *i.e.*, How2comm achieves a better performance-bandwidth trade-off with spatial-channel filtering than Where2comm. **(ii)** Moreover, our framework seeks comparable performances as the SOTAs by consuming less bandwidth. The noteworthy improvements demonstrate that the proposed communication mechanism filters invalid semantics and maintains performance by mutual information maximization.

**Robustness to Localization and Heading Errors.** We verify the detection performance of How2comm under varying pose errors of collaborators in Figures 5 and 6 following the noise settings in [40]. Specifically, the localization and heading errors are sampled from Gaussian distributions with a standard deviation of $\sigma_{xyz} \in [0, 0.5]$ m and $[0°, 1.0°]$, respectively. As shown in the figures, the performance of all intermediate collaboration models consistently deteriorates due to feature map misalignment as the pose errors increase. Noticeably, How2comm is superior to the previous SOTA models and No Fusion across three datasets under all error levels, while some models (*e.g.*, V2VNet and When2com) are even weaker than No Fusion when the error exceeds 0.2 m and 0.2° on the DAIR-V2X dataset. This comparison demonstrates the robustness of How2comm against collaboration pose noises. One reasonable explanation is that our framework captures perceptually critical and holistic information across heterogeneous agents via the tailored STCFormer.

**Robustness to Transmission Delay.** As noted in Figure 1(b), temporal asynchrony due to transmission delay results in two-sided fusion errors and harms the collaboration procedure. We analyze the sensitivity of existing models to varying delays (*i.e.*, from 0 to 400 ms) in Figure 7. Noticeably, all the intermediate fusion methods inevitably degrade with increasing transmission delay due to misleading feature matching. Nevertheless, How2comm maintains higher precision than SOTAs at all delay levels across all three datasets and improves AP@0.7 by 7.5% than No Fusion on DAIR-V2X under a severe delay (400 ms). This robustness to the transmission delay proves that How2comm accomplishes temporal alignment of two-sided features by predicting the collaborators' future features.

## 4.3 Ablation Studies

We perform thorough ablation studies on all datasets to understand the necessity of the different designs and strategies in How2comm. Table 2 shows the following vital observations.

**Rationality of Communication Mechanism.** **(i)** The spatial and channel attention queries are first removed separately to perform incomplete message filtering. The decreased performance implies that both query patterns contribute to sharing sparse yet salient features among agents. **(ii)** There is a significant degradation in the detection results on all datasets when the communication mechanism lacks mutual information supervision. A plausible deduction is that our supervision mitigates the loss of valuable information due to spatial-channel feature filtering.

**Effect of Delay Compensation Strategy. (i)** Here, we remove the scale matrix to verify its effect. The poor results show that appropriate scaling of predicted features promotes effective temporal alignment. **(ii)** Furthermore, the performance drop caused by the self-supervised training removal shows the importance of imposing motion representation supervision for collaborator-shared features.

Table 2: Ablation study results of the proposed components and strategies on all the datasets. "w/o" stands for the without.

| Components/Strategies | DAIR-V2X AP@0.5/0.7 | V2XSet AP@0.5/0.7 | OPV2V AP@0.5/0.7 |
|---|---|---|---|
| **Full Framework** | **62.36/47.18** | **84.05/67.01** | **85.42/72.24** |
| *Rationality of Communication Mechanism* | | | |
| w/o Spatial Query | 61.25/46.33 | 83.14/66.15 | 84.52/71.40 |
| w/o Channel Query | 61.76/46.52 | 83.70/66.58 | 85.23/71.76 |
| w/o Mutual Information | 60.61/46.04 | 82.53/65.75 | 84.06/70.69 |
| *Effect of Delay Compensation Strategy* | | | |
| w/o Scale Matrix | 62.17/46.86 | 83.61/66.63 | 85.08/71.54 |
| w/o Self-supervised Training | 60.55/45.77 | 82.84/65.78 | 84.20/70.36 |
| *Importance of STCFormer* | | | |
| w/o Temporal Cross-Attention | 60.13/45.92 | 83.09/65.88 | 84.27/70.85 |
| w/o Exclusive Spatial Attention | 59.46/45.06 | 82.65/65.70 | 83.81/71.19 |
| w/o Common Spatial Attention | 61.24/46.63 | 83.48/66.41 | 84.39/71.44 |
| w/o Adaptive Late Fusion | 61.93/46.77 | 83.72/66.69 | 84.86/71.64 |
| *Impact of Keypoint Number $N_v$* | | | |
| $N_v = 6$ | 61.68/46.05 | 82.82/66.15 | 84.67/71.38 |
| $N_v = 9$ (Default) | 62.36/47.18 | 84.05/67.01 | 85.42/72.24 |
| $N_v = 12$ | 62.13/46.57 | 84.12/66.59 | 85.28/72.01 |
| *Necessity of Decoupled Design* | | | |
| w/o Decoupled Design | 60.08/45.36 | 82.41/65.38 | 84.02/71.15 |

**Importance of STCFormer.** STCFormer is evaluated in three dimensions. **(i)** We first find that temporal cross-attention provides beneficial gains due to performance deterioration when discarded. It is because the meaningful temporal clues in historical frames bridge the single-frame detection gap. **(ii)** Then, exclusive and common spatial attention modules are removed separately to explore the impact on performance. The consistently decreased results on each dataset suggest that integrating distinct spatial semantics is indispensable for pragmatic collaboration. **(iii)** Finally, the adaptive late fusion is replaced by pixel-wise addition. The gain decline suggests that our fusion paradigm provides new insights for aggregating the perceptual advantages of distinct spatial semantics.

**Impact of Keypoint Number $N_v$.** Empirically, we set the variable number of keypoints for the decoupled spatial attention modules to perform experiments and find that 9 keypoints achieve the most competitive detection performance. Conversely, too few keypoints may cause valuable semantics loss and too many keypoints may cause performance bottlenecks due to accumulated errors. The above finding inspires us to determine the appropriate keypoint number that samples more rich visual clues and captures more pragmatic collaboration information from collaborators.

**Necessity of Decoupled Design.** Ultimately, we set $\mathcal{E}_j^t$ and $\mathcal{C}_j^t$ to 1 while maintaining two spatial attention branches in the STCFormer to justify the decoupled design. Without explicitly defining exclusive and common features, spatial attention would indiscriminately sample the entire region due to the lack of guidance from prior perception information, which may introduce excessive collaborator noises and impede bridging to collaboration heterogeneity.

## 4.4 Qualitative Evaluation

**Visualization of Detection Results.** To illustrate the perception performance of different models, Figure 8 shows the detection visualizations of two challenging scenarios from the DAIR-V2X

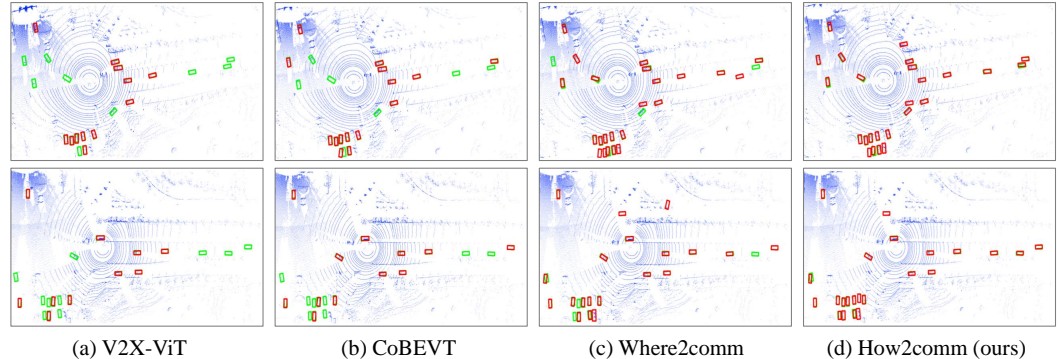

| (a) V2X-ViT | (b) CoBEVT | (c) Where2comm | (d) How2comm (ours) |

Figure 8: Detection visualization comparison in real-world scenarios from the DAIR-V2X dataset. Green and red boxes denote the ground truths and detection results, respectively.

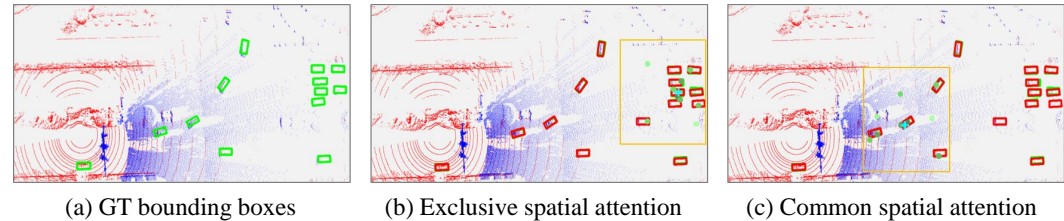

| (a) GT bounding boxes | (b) Exclusive spatial attention | (c) Common spatial attention |

Figure 9: Visualization of learned exclusive and common spatial attention maps. The red and blue point clouds are derived from the ego vehicle and infrastructure, respectively. In (b)&(c), cyan cross marker denotes the target query in the deformable cross-attention. The 9 sampled keypoints are represented by the forestgreen dots whose colors reflect the corresponding attention weights.

dataset [52] under default noise settings. How2comm axiomatically achieves more robust and accurate detection compared to previous SOTA models, including V2X-ViT [40], CoBEVT [38], and Where2comm [7]. Concretely, our method produces more predicted bounding boxes well aligned with the ground truths. The merits may lie in two aspects. **(i)** Our delay compensation strategy mitigates feature misalignment due to temporal asynchrony, improving detection performance. **(ii)** The proposed STCFormer provides effective temporal context clues for detecting fast-moving objects and fuses meaningful information from nearby agents to compensate for the occluded perspective.

**Visualization of Spatial Attention Maps.** We show visualizations of exclusive and common spatial attention (ESA/CSA) in Figure 9 to justify the effectiveness of our feature decoupling philosophy. **(i)** Intuitively, from Figure 9(b), ESA effectively samples keypoints at the exclusive perception region from the infrastructure, providing complementary information for the ego agent to promote perception ability. **(ii)** In Figure 9(c), CSA mitigates the detection gap due to collaboration heterogeneity by sampling keypoints that reasonably focus on the common perception region between two agents.

## 5 Conclusion and Limitation

This paper presents How2comm, a novel collaborative perception framework to tackle existing issues jointly. How2comm maximizes the beneficial semantics in filtered features and accomplishes temporal alignment via the feature flow estimation. Moreover, our STCFormer holistically aggregates the spatial semantics and temporal dynamics among agents. Extensive experiments on several multi-agent datasets show the effectiveness of How2comm and the necessity of all its components.

**Limitation and Future Work.** The current work only exploits the short-term historical frames. In future work, we plan to expand the utilization of temporal information to long-term point cloud sequences. Also, we will explore optimizing the feature flow prediction with uncertainty estimation.

**Acknowledgment.** This work is supported in part by the National Key R&D Program of China under Grant 2021ZD0113503 and in part by the Shanghai Municipal Science and Technology Major Project under Grant 2021SHZDZX0103.

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
