# How2comm: Communication-Efficient and Collaboration-Pragmatic Multi-Agent Perception

**Dingkang Yang**[1,2†]     **Kun Yang**[1†]     **Yuzheng Wang**[1]     **Jing Liu**[1]
**Zhi Xu**[1]     **Rongbin Yin**[5]     **Peng Zhai**[1,2∗]     **Lihua Zhang**[1,2,3,4∗]

[1]Academy for Engineering and Technology, Fudan University
[2]Cognition and Intelligent Technology Laboratory (CIT Lab)
[3]Engineering Research Center of AI and Robotics, Ministry of Education
[4]AI and Unmanned Systems Engineering Research Center of Jilin Province
[5]FAW (Nanjing) Technology Development Company Ltd
{dkyang20, kungyang20, pzhai, lihuazhang}@fudan.edu.cn

## 1   Highlights of Our Contribution

We propose *How2comm*, a collaborative perception framework that seeks a trade-off between perception performance and communication bandwidth. The contributions of our How2comm are highlighted as follows:

- We propose a Mutual Information-aware Communication (MIC) mechanism to address the **communication redundancy** issue. On the one hand, MIC directs collaborators to share spatial-channel sparse, yet perceptually critical information through spatial-channel message filtering. On the other hand, the mutual information maximization supervision is introduced to mitigate the depletion of locally valuable semantics in the transmitted features.

- We design a Flow-guided Delay Compensation (FDC) strategy to address the **transmission delay** issue. FDC eliminates two-sided fusion error at the feature level due to the temporal asynchrony by predicting the future features of the collaborators. FDC provides a promising solution to mitigate the inevitable transmission delay in collaborative perception by a self-supervised motion estimation paradigm.

- We present a Spatio-Temporal Collaboration transFormer (STCFormer) to address the **collaboration heterogeneity** issue. STCFormer delivers a unified structure for pragmatic and robust collaboration by simultaneously modelling the temporal dynamics and spatial semantics among heterogeneous agents.

## 2   Details about the System Pipeline

Alg. 1 presents the system pipeline of the proposed multi-agent collaborative perception framework How2comm. We assume that there is a data transmission delay $\tau$ (*i.e.*, $t = t_0 + \tau$) between the ego agent ($i$) and the collaborators. The point clouds $\{\boldsymbol{X}_j^{t_0-k}, ..., \boldsymbol{X}_j^{t_0} \mid 1 \leq j \leq N, j \neq i\}$ of $N-1$ collaborators and the ego agent's historical frames $\{\boldsymbol{X}_i^{t_0}, ..., \boldsymbol{X}_i^t\}$ are defined as the system inputs, which are synchronized to the current coordinate system of the ego agent. As stated in the *main paper*, the primary goal of How2comm is to maximize the 3D object detection performance of the ego agent by aggregating the semantic information from the above system inputs under a communication budget $B$. Specifically, **FFN** is the feed-forward network, **TCA**, **ESA**, **CSA**, and **ALF** denote the

---

[†]Equal contributions.
[∗]Corresponding authors.

37th Conference on Neural Information Processing Systems (NeurIPS 2023).

**Algorithm 1:** System pipeline of How2comm

---

**Input** : $\{\boldsymbol{X}_j^{t_0-k}, ..., \boldsymbol{X}_j^{t_0} \mid 1 \le j \le N, j \ne i\}$, $\{\boldsymbol{X}_i^{t_0}, ..., \boldsymbol{X}_i^t \mid t = t_0 + \tau\}$

1   # Each agent processes independently
2   # 1. Mutual Information-aware Communication
3   **for** *each agent* $j = 1, 2, ..., N$ **do**
4      $\boldsymbol{F}_j^{t_0} = f_{enc}(\boldsymbol{X}_j^{t_0}) \in \mathbb{R}^{H \times W \times C}$;
5      Obtain attention queries: $\mathcal{A}_{i,s}^{t_0} \in \mathbb{R}^{H \times W \times 1}$, $\mathcal{A}_{i,c}^{t_0} \in \mathbb{R}^{1 \times 1 \times C}$;
6      **if** $j == i$ **then**
7         Broadcast request queries: $\mathcal{R}_{i,s}^{t_0} = 1 - \mathcal{A}_{i,s}^{t_0}$, $\mathcal{R}_{i,c}^{t_0} = 1 - \mathcal{A}_{i,c}^{t_0}$;
8      **else**
9         $\mathcal{M}_j^{t_0} = $ **Eq. (2)\*** $\in \{0, 1\}^{H \times W \times C}$;
10        $\mathcal{O}_j^{t_0}, \mathcal{S}_j^{t_0} = f_{flow}([\boldsymbol{F}_j^{t_0-k}, ..., \boldsymbol{F}_j^{t_0}])$;
11        Send $\tilde{\boldsymbol{F}}_j^{t_0} = \boldsymbol{F}_j^{t_0} \odot \mathcal{M}_j^{t_0}$ and $\{\mathcal{O}_j^{t_0}, \mathcal{S}_j^{t_0}\}$ to the $i$-th agent;

12   # For ego agent
13   # 2. Flow-guided Delay Compensation
14   $\boldsymbol{Z}_j^t = f_{warp}(\tilde{\boldsymbol{F}}_j^{t_0}, (t - t_0) \cdot \mathcal{O}_j^{t_0}) \odot \mathcal{S}_j^{t_0}$;
15   # 3. Spatio-Temporal Collaboration Transformer
16   $\mathcal{I}_i^t / \mathcal{I}_j^t = \sigma(\varphi_m(f_{gen}(\boldsymbol{F}_i^t / \boldsymbol{Z}_j^t))) \in [0, 1]^{H \times W}$;
17   $\boldsymbol{Z}_{j,e}^t = f_{sel}((1 - \mathcal{I}_i^t) \odot \mathcal{I}_j^t) \odot \boldsymbol{Z}_j^t$, $\boldsymbol{Z}_{j,c}^t = f_{sel}(\mathcal{I}_i^t \odot \mathcal{I}_j^t) \odot \boldsymbol{Z}_j^t$;
18   $\boldsymbol{H}_i^t = \textbf{TCA}([\tilde{\boldsymbol{F}}_j^{t_0}, \boldsymbol{F}_i^{t_0}, ..., \boldsymbol{F}_i^t])$;
19   $\boldsymbol{F}_{i,e}^t = \textbf{ESA}(\boldsymbol{H}_i^t, \{\boldsymbol{Z}_{j,e}^t \mid 1 \le j \le N, j \ne i\}) + \boldsymbol{H}_i^t$, $\boldsymbol{F}_{i,e}^t = \textbf{FFN}(\boldsymbol{F}_{i,e}^t) + \boldsymbol{F}_{i,e}^t$;
20   $\boldsymbol{F}_{i,c}^t = \textbf{CSA}(\boldsymbol{H}_i^t, \{\boldsymbol{Z}_{j,c}^t \mid 1 \le j \le N, j \ne i\}) + \boldsymbol{H}_i^t$, $\boldsymbol{F}_{i,c}^t = \textbf{FFN}(\boldsymbol{F}_{i,c}^t) + \boldsymbol{F}_{i,c}^t$;
21   $\mathcal{F}_i^t = \textbf{ALF}(\boldsymbol{F}_{i,e}^t, \boldsymbol{F}_{i,c}^t) \in \mathbb{R}^{H \times W \times C}$;
22   # 4. Detection Decoders
23   $\mathcal{Y}_{i,r}^t = f_{dec}^r(\mathcal{F}_i^t) \in \mathbb{R}^{H \times W \times 7}$, $\mathcal{Y}_{i,c}^t = f_{dec}^c(\mathcal{F}_i^t) \in \mathbb{R}^{H \times W \times 2}$.

---

proposed temporal cross-attention, exclusive spatial attention, common spatial attention, and adaptive late fusion components, respectively. **Eq.** $(\cdot)$**\*** represents the equation in the *main paper*, $\sigma$ and $\varphi_m(\cdot)$ denote the sigmoid activation and max pooling function, respectively. $\odot$ and $\cdot$ denote the point-wise multiplication and scalar multiplication, respectively. As Alg. 1 shows, How2comm leverages two tailored components to facilitate an efficient communication pattern and eliminate temporal asynchrony, respectively. Moreover, the proposed STCFormer component effectively aggregates the historical context clues and comprehensive collaborator information, leading to a more pragmatic collaboration. The output prediction results of the ego agent are produced based on the refined visual representation $\mathcal{F}_i^t$. Note that we present the processing of each agent in an iteration format only for clarity; these processes are conducted in parallel during the implementation.

## 3   Experiments

### 3.1   Implementation Details

The height and width resolution of the feature encoder $f_{enc}(\cdot)$ is set to 0.4 m. The extracted BEV feature map size is $(100, 252, 64)$ on the DAIR-V2X [11] dataset and $(100, 352, 64)$ on the V2XSet [9] and OPV2V [10] datasets.

**Communication Volume.** As stated in the *main paper*, the $j$-th agent, as a collaborator, employs the learned binary message filtering matrix $\mathcal{M}_j^{t_0} \in \{0, 1\}^{H \times W \times C}$ to obtain the sparse feature as $\tilde{\boldsymbol{F}}_j^{t_0} = \boldsymbol{F}_j^{t_0} \odot \mathcal{M}_j^{t_0} \in \mathbb{R}^{H \times W \times C}$ and sends $\tilde{\boldsymbol{F}}_j^{t_0}$ to the ego agent. Moreover, the feature flow $\mathcal{O}_j^{t_0} \in \mathbb{R}^{H \times W \times 2}$ and scale matrix $\mathcal{S}_j^{t_0} \in \mathbb{R}^{H \times W \times 1}$ are also transmitted. We count the message size by byte in the log scale with base 2 by formulating the communication volume as follows:

$$\log_2[(|\mathcal{M}_j^{t_0}| + 3HW) \times 32/8], \tag{1}$$

where $|\cdot|$ is the $L0$-norm operation counting non-zero elements. 32 represents that each element of the transmitted feature is the float32 data type, and 8 represents that byte is used as the metric. The vanilla calculation format of communication volume as [1] is applied for other feature compression or spatial filtering methods.

**Bandwidth Limitation.** The existing commercial V2V communication products based on 802.11p achieve transmission rates close to 6 MBps, while LTE-V2V channels can reach approximately 10 MBps [5]. Given the capturing frequency of 10 Hz for commonly used LiDAR [10], LTE-V2V communication products can transmit message sizes of about 1 MB in 100 ms. Therefore, we set a basic bandwidth limitation of $\approx 1$ MB to simulate the restricted communication resources in real V2X/V scenarios and evaluate the performance of all models under this limitation setting.

### 3.2 Experimental Setup

**Communication Graph.** Following the settings of existing works [3, 4, 6, 9, 10], we identify an agent (*e.g.*, vehicle) as the ego agent and set other agents within the communication range as collaborators during the collaboration. The communication range is set to 70 m [1] in practice. With this design, our primary goal is to improve the perception ability of the ego agent by aggregating complementary information from its local historical observations and collaborator-shared features. In practice, the ego agent can broadcast its metadata (*e.g.*, poses, extrinsic, and sensor type) using the existing Vehicle-to-Vehicle/Everything communication channel. The transmission delay of the metadata is negligible due to the small message size. Upon receiving the metadata of the ego agent, the collaborators will first project their local point cloud observations under the ego coordinate system through the coordinate transformation. Moreover, to extract historical context clues efficiently, the preceding point cloud frames considered in our framework are synchronized to the current coordinate frame via the ego-motion compensation module [7].

**Feature Encoder.** For optimizing processing latency and memory usage, we employ the anchor-based method PointPillar [2] as the feature encoder $f_{enc}(\cdot)$ to extract visual representations from the captured point cloud frames. The encoder converts the raw point clouds into the stacked pillars and scatters them into a 2D pseudo-image. A feature pyramid network processes the pseudo-image and outputs the final Bird's-Eye-View (BEV) features. Given the point cloud $\boldsymbol{X}_i^t$ of $i$-th agent at timestamp $t$, the corresponding extracted BEV feature map is $\boldsymbol{F}_i^t = f_{enc}(\boldsymbol{X}_i^t) \in \mathbb{R}^{H \times W \times C}$, where $H, W, C$ stand for the height, width, and channel. The network parameters of the feature encoder $f_{enc}(\cdot)$ are shared among all agents.

**Selection Function.** As shown in Eq. (2) in *the main paper*, the selection function $f_{sel}(\cdot)$ is introduced to select the salient spatial-channel elements and obtain the binary message filtering matrix $\mathcal{M}_j^{t_0}$. Specifically, the elements of the input matrix are first arranged in descending order. Considering the dynamic communication resources of the realistic V2V/X channel, $f_{sel}(\cdot)$ leverages the given bandwidth limitation to set a communication threshold, which determines the spatial-channel elements that will be transmitted. The matrix $\mathcal{M}_j^{t_0}$ is obtained by a filtering operation, where the corresponding elements in the input matrix greater than the communication threshold are set to 1 and 0 verses.

**Time Embedding.** Time embedding is presented to explicitly introduce the temporal relationships of input BEV features into the temporal cross-attention module. Following [12], we apply a linear layer to encode the relative time $[t - t_0, t - t_0, ..., 0] \in \mathbb{R}^{1 \times (N\tau+1)}$ of the input features $[\bar{\boldsymbol{F}}_j^{t_0}, \boldsymbol{F}_i^{t_0}, ..., \boldsymbol{F}_i^t]$ as a time embedding, whose shape is $(N\tau + 1, 1, 1, C)$. Then, the input features are reshaped to the size of $(N\tau + 1, H, W, C)$ and added with the learned time embedding across all spatial locations.

### 3.3 Qualitative Evaluation

Figure 1 shows more detection results to verify the effectiveness of How2comm in high-density traffic flow scenarios. We show the visual comparisons between How2comm and current SOTA methods, including V2X-ViT [9], CoBEVT [8], and Where2comm [1], under the collaboration noises and communication bandwidth constraints aligned with the *main paper*. Unsurprisingly, How2comm has fewer regression misalignments and fewer undetected objects, resulting in a more robust performance. In comparison, previous methods fail to tackle fast-moving objects and perceive remotely occluded regions. The indisputable evidence is that some ground truth bounding boxes are not well matched with their detection results.

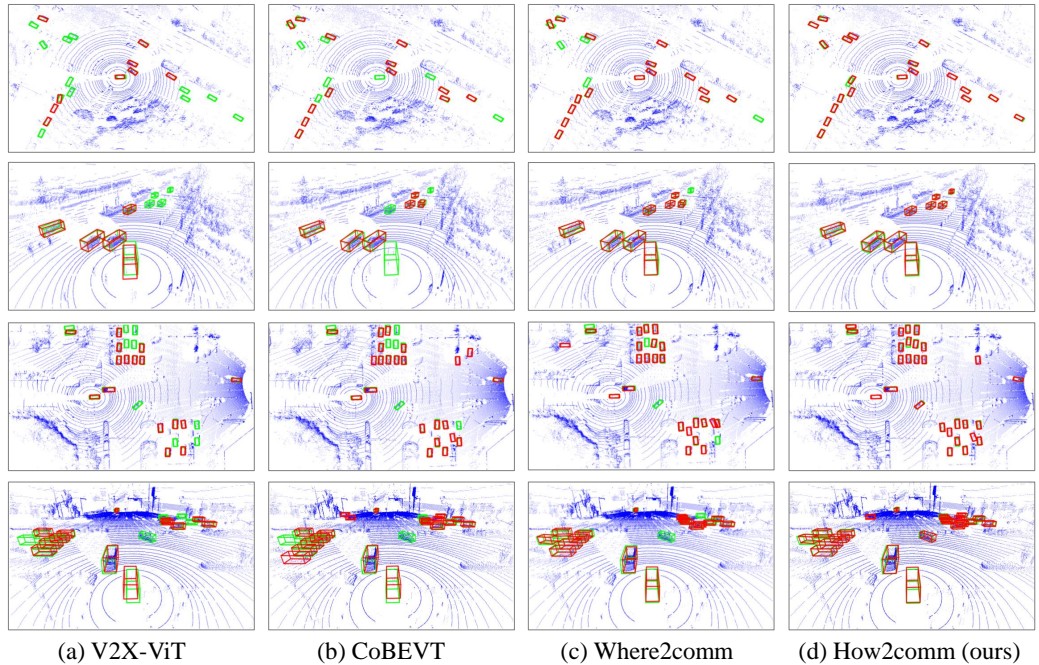

(a) V2X-ViT        (b) CoBEVT        (c) Where2comm        (d) How2comm (ours)

Figure 1: Detection visualization comparison in real-world scenarios. Green and red boxes denote the ground truth and detection results, respectively.

## 4 Broader Impacts

▶ The positive impact of this work is that How2comm can leverage vehicular communication technology to facilitate information sharing and enhance the vehicles' perception ability of the surrounding environment, reducing the incidence of traffic accidents.

▶ Additionally, the collaborative perception ability provided by How2comm can be used to monitor real-time traffic conditions and improve traffic efficiency.

▶ Collaborative perception systems may leak vehicle-related information (*e.g.*, locations and driving speed), violating the owner's privacy. Meanwhile, illegal personnel may invade the vehicle communication channel to attack the collaborative perception system, leading to the vehicles' misjudgment of the surrounding environment or malicious remote control.

▶ In summary, this work facilitates Vehicle-to-Vehicle/Everything (V2V/X)-oriented multi-agent communication applications in autonomous driving.

## 5 Discussion on the Realistic Limitations

As an emerging field, there are many challenges in building a robust multi-agent collaborative perception system. This paper addresses three of the biggest challenges, including communication redundancy, transmission delay, and collaboration heterogeneity, through tailored components. Although previous works have appropriately focused on these dilemmas, collaborative perception systems still have considerable room for improvements in the trade-off between perception performance and communication bandwidth. How2comm achieves superior performance with lower communication overheads in real V2V/X scenarios with limited communication capacity. We further discuss other realistic limitations and indicate future research directions to optimize our system.

▶ For the attack issue, How2comm could improve the robustness against potential adversarial attacks by introducing an adversarial training paradigm. In addition, by focusing on decoupled spatial regions and discovering perceptually critical information, How2comm is relatively less likely to be attacked.

▶ For the data bias issue, How2comm has demonstrated its preliminary generalization on real-world and simulated datasets with different configurations and sensor facilities. In future work, How2comm

could disentangle causal relationships among variables in the collaborative perception pipeline with the assistance of causal-effect techniques, further enhancing its ability to generalize to more complex real-world scenarios.

▶ For the domain gap issue, How2comm could hopefully achieve sim2real adaptation and inter-agent adaptation by introducing an unsupervised sim2real domain bridging mechanism.