# OpenReview forum: "How2comm: Communication-Efficient and Collaboration-Pragmatic Multi-Agent Perception"
_NeurIPS.cc/2023/Conference — NeurIPS 2023 poster_

### Official Review · Reviewer_kdrk · 2023-06-26

**Soundness:** 4 excellent
**Presentation:** 4 excellent
**Contribution:** 4 excellent
**Rating:** 8
**Confidence:** 5

**Summary:**

This article proposes a communication-efficient and collaboration-pragmatic framework that effectively reduces object detection errors caused by transmission delays in multi-agent collaborative perception tasks, while achieving a balance between bandwidth utilization and detection accuracy. This paper introduces the spatial-channel message filter to assist collaborators in selecting important feature information. Additionally, to tackle the issue of information transmission delays, it employs flow-guided delay compensation operations to predict the future state of collaborators, thereby reducing errors caused by information transmission latency. Finally, this article integrates feature information into the STCFomer framework and leverages two attention mechanism modules to effectively improve the accuracy of target detection after feature fusion.

**Strengths:**


1. The article proposes an effective solution to one of the primary issues in multi-agent collaborative perception tasks, achieving a reduction in detection errors after feature fusion while maintaining a balance between bandwidth utilization and detection accuracy. The proposed method demonstrates improved detection accuracy on three main datasets.
2. The designed STCFormer architecture in this article effectively integrates the information transmitted to the ego agent. By leveraging temporal cross-attention and exclusive spatial attention modules, it enhances information in both the temporal and spatial dimensions.
3. The article exhibits fluency in language, completeness in experiments, and reliable comparative results. It has the potential to contribute to the development of the research field.

**Weaknesses:**

The paper's ablation experiments are expected to provide more detailed analysis and discussion, including jointly eliminating several modules. For example, simultaneously removing the delay compensation strategy.

**Questions:**

1. In Section 3.4 "Flow-guided Delay Compensation", in line 150, should the referenced image be Figure 2 rather than Figure 1?
2. In Section 3.3 "Spatial-channel message filtering", in line 128, Why is the feature sent to other agents required to undergo the "1-A" operation?
3. If the range and content of point clouds collected by different agents may overlap but are not entirely consistent, how can the method ensure that the queries sent by the ego agent effectively associate with the same vehicles in the feature map of collaborators?
4. Because the chosen dataset for this article does not provide point cloud flows, I would like to ask if the authors, when performing the "flow-guided delay compensation" operation, treated the continuous frame sequences in the dataset as a whole and re-segmented it? It would be helpful if the authors could explain the data processing steps involved.


**Limitations:**

The authors have acknowledged the limitation of their work in only being able to handle short-term historical frames. However, this is influenced by various factors such as the dataset. The authors have made a comprehensive effort within the existing scope of their work.

---

> ### Author Rebuttal · Authors · 2023-08-10
>
> We thank the reviewer for recognizing our effective solution, complete experiments, and reliable results. Below are some specific responses. The common ablation experiments can be found in the global response.
>
> **Q1**:  Reference problem for the figure in line 150.
>
> **A1**:  Thank you for the conscientious checking! We will correct this typo in the revision.
> ***
> **Q2**:  Explanation of the $1-A_{i, s/c}^{t_0}$ operation.
>
> **A2**:  As stated in lines 122-126 of the Main Paper, the obtained spatial and channel queries $A_{i, s/c}^{t_0}$ reflect salient spatial locations and semantically meaningful channels in the feature $F_{i}^{t_0}$. As stated in lines 90-91 of the Supplementary, the two request queries $R_{i, s/c}^{t_0}$ obtained via operation $1-A_{i, s/c}^{t_0}$ reflect potentially unconfident positions in the ego agent due to practical difficulties such as occlusion, which require complementary information to improve detection accuracy. To this end, collaborators can produce a spatial-channel binary filtering matrix to assist in transmitting spatial-channel sparse, yet perceptually critical features based on the needs reflected in the request queries and their own strengths.
> ***
> **Q3**:  Discussion of the position association.
>
> **A3**:  We follow the existing works using point cloud projection and coordinate system transformation to synchronize the collaborator feature maps to the ego agent's coordinate frame. Based on this, the queries learned based on the ego agent feature are spatially aligned with the collaborator feature maps to support subsequent cross-agent interaction and collaboration. Concretely, as stated in lines 108-110 of the Main Paper, we assume that the collaborators project their own point cloud data to the ego agent's coordinate system upon receiving the ego agent's metadata (e.g., poses and extrinsic) before feature extraction. Also, we utilize the differential transformation and sampling operator provided by V2X-ViT [1] to transform the collaborator feature maps to the current ego agent's pose.
>
> [1] Xu R, Xiang H, Tu Z, Xia X, Yang M-H, Ma J, V2X-ViT: Vehicle-to-Everything Cooperative Perception with Vision Transformer, European Conference on Computer Vision 2022.
> ***
> **Q4**: Discussion of the data processing steps.
>
> **A4**: In practice, we optimize the data processing steps of existing codebases to return consecutive frame sequences with configurable lengths.
>
> The complete raw data in existing datasets consists of multiple scenes, where each scene has a continuous frame sequence with a fixed length. The current collaborative perception codebase treats sensor data from multiple agents at the same timestamp as one data sample and assigns a unique sample index. In addition, the scene index is also provided to reflect the scene in which the data sample is located. When loading data, the training script provides the sample indexes in a disrupted order and loads the corresponding data samples. This data-loading step limits the usage of historical samples.
>
> To this end, we provide a new data processing step: when loading a target data sample with sample index $n$, we simultaneously load multiple previous historical data samples, such as $( n-2, n-1 )$, constituting a sequence of consecutive frames. It is worth noting that we utilize the scene index to determine whether the loaded samples belong to the same scene to avoid data loading errors. As described in lines 150-152 of the Main Paper, the loaded sequence of consecutive frames is passed into the generator to extract temporal information and output the feature flow and scale matrix.
>
> If this paper is accepted, we will release the complete codebase for community development.
> ***
> **Q5**: About limitations.
>
> **A5**: The authors are very grateful to the reviewer for helping us clarify that many practical limitations are indeed impacted by diverse factors, especially the current dataset configuration and format. As a complement, we have further discussed the broad impacts and realistic limitations of How2comm in Sec. 5 and 6 of the Supplementary, indicating future directions for optimizing our system.

---

> > ### Comment · Reviewer_kdrk · 2023-08-17
> >
> > Thank you for the author's answer. I am looking forward to the smooth publication of the paper. I hope the author can do a good job in open sourcing to promote the development of the field.

---

> > > ### Author Response · Authors · 2023-08-17
> > > **Official Comment by Authors to Reviewer kdrk**
> > >
> > > Well, no exaggeration; it was an honor to meet you！
> > >
> > > We will follow your constructive suggestions in the released version to contribute to the development of the community.

---

### Official Review · Reviewer_ToHv · 2023-06-30

**Soundness:** 3 good
**Presentation:** 3 good
**Contribution:** 3 good
**Rating:** 6
**Confidence:** 4

**Summary:**

This paper presents a collaborative perception system called How2comm, which aims to address three common challenges encountered in the collaborative perception setting:

1) Communication redundancy: To mitigate communication redundancy, the paper introduces a Mutual Information-aware communication approach. This approach incorporates a spatial-channel message-filtering strategy and a supervision signal that seeks to maximize the mutual information between the transmitted features and the original features.

2) Transmission delay: To tackle transmission delay, the paper proposes the utilization of flow generation and warping techniques for effective delay compensation.

3) Collaboration heterogeneity: To handle collaboration heterogeneity, the paper employs a carefully designed decoupled spatial attention mechanism in the STC transformer. This mechanism enables the network to learn exclusive and common collaborator features. Additionally, the paper introduces adaptive late fusion to combine these features in a meaningful manner.

By implementing these approaches, the proposed system aims to enhance collaboration in the perception setting by addressing communication redundancy, transmission delay, and collaboration heterogeneity.

**Strengths:**

1) This paper presents a system that addresses three significant challenges in collaborative perception simultaneously, making it a relatively complex undertaking. However, it is worth noting that the paper is exceptionally well-organized, and the writing surpasses the average standard in terms of clarity and coherence.

2) Regarding the first two challenges, namely communication redundancy and transmission delay, the proposed designs exhibit both originality and effectiveness. Particularly noteworthy is the intuitive flow-based compensation method, which proves to be highly effective in practice.

3) The experimental results presented in the paper are promising, and the authors provide a clear and comprehensive demonstration of their findings. This aspect enhances the overall credibility of the research and its potential practical implications.

**Weaknesses:**

The third challenge, collaboration heterogeneity, raises some concerns and requires further clarification and evidence of the authors' approach to addressing it.

1) From the reviewer's perspective, collaboration heterogeneity primarily refers to the heterogeneity of sensors, which may result in heterogeneity within the feature space. For instance, when one agent uses a camera while another employs a LiDAR sensor, additional operations are needed to ensure proper fusion of the information. Even in Figure 1(d), the dominant cause of collaboration heterogeneity is the differing beam numbers of the two LiDAR sensors. It is mentioned that there might be a misunderstanding on the reviewer's part, as a clear definition of collaboration heterogeneity is not evident from lines 42 to 52 of the paper.

2) In Section 3.5, it is observed that the network architectures for the feature fusion module of exclusive and common features appear to be identical. However, in lines 48 to 50, the authors indicate that the fusion of the common area is the more challenging aspect. This raises the expectation among readers for specialized designs addressing the fusion of common features. Have the authors considered the heterogeneity in the features of different collaborators?

3) Table 2 displays the ablation study of excluding Exclusive Spatial Attention and Common Spatial Attention. It would be helpful to know if the authors have also conducted experiments without explicitly defining exclusive or common features, using the same attention module structure twice. In this scenario, both $\mathcal{E}$ and $\mathcal{C}$ would be set to 1 throughout the dataset. This question arises due to the concerns expressed in the second point regarding the efficiency of the decoupled design.

Given these uncertainties, it is crucial to provide additional explanations and experimental results to enhance the understanding of collaboration heterogeneity as perceived by the authors and the logical efficiency of the STCFormer in overcoming this challenge.

**Questions:**

The questions are almost all related to the [Weaknesses] section:
1. A clear definition of Collaboration Heterogeneity is needed;

2. The design logic of the common feature attention module, and why it can tackle the difficulty mentioned in the introduction;

3. Missing ablation study for STCFormer.

4. Since it is a comparatively complex system and latency is one of the biggest issues in collaborative perception systems, can authors provide some inference time comparison with other SOTA methods? such as where2comm.

Minor questions:

1. In the ablation study (Table 2), it seems that the dynamic late fusion only has a very marginal improvement in performance. Is any explanation for that?

2. For the dots in Figure 8 (b)(c), does the lighter dot mean that the weight is lower or higher?

3. In L165-L167, the expression of "the decoupled spatial semantics and temporal dynamics among agents" is a little bit confusing. The STCFormer seems only to decouple the spatial semantics into common and exclusive, then why "temporal dynamics" is here?



**Limitations:**

Another limitation of this paper is that it assumes the features from different agents' sensors are readily available for fusion when designing the STCFormer, despite acknowledging the presence of collaboration heterogeneity. This assumption may oversimplify the challenges associated with fusing heterogeneous sensor data and overlook the potential difficulties in achieving seamless integration of diverse sensor modalities. The paper could further explore and address the practical issues and complexities related to the fusion of sensor data with varying characteristics and formats, providing a more comprehensive and robust solution for collaboration heterogeneity in the context of collaborative perception systems.

---

> ### Author Rebuttal · Authors · 2023-08-10
>
> We thank the reviewer for recognizing our effective designs and comprehensive experiments. Below are some specific responses. The common questions regarding the definition of collaboration heterogeneity, ablation studies, and inference time can be found in the global response.
>
> **Q1**:  Design logic of the common spatial attention (CSA).
>
> **A1**:  We want to clarify that in lines 46-50, we use Fig. 1(d) to introduce the exclusive and common perception regions, and to explain that it is essential to jointly fuse the valuable spatial semantics of these two types of regions. The original intent of the description was not to imply that the fusion of common area features is more challenging. We apologize for the misunderstanding and will improve the presentation in the revision.
>
> Next, we introduce the design logic of the CSA module in three parts:
> *  As stated in lines 182-185, we get the common candidate map following $\mathcal{C}_j^t = \mathcal{I}_i^t \odot \mathcal{I}_j^t$. Since the importance map reflects the perceptual critical levels of features at each spatial location, $\mathcal{C}_j^t$ reflects the regions where both the ego agent and the collaborator have high importance, i.e., common perception regions.
> *  We use type-dependent linear layers to project agent features with distribution discrepancies to the same feature space based on the agent types. Such type-dependent operations would help capture the agent heterogeneity and optimize feature fusion. Since the common perception region reflects the heterogeneous characterizations of the same spatial semantics from the ego agent and collaborator, the type-dependent linear layers would learn the data characteristics of distinct agents and thus bridge the data discrepancies in the CSA module.
> *  As stated in lines 193-195, we introduce a learnable sampling method to enhance the target queries' feature. This method can adaptively select keypoints and assign attention weights based on feature characteristics, which can learn to set reasonable weights based on agent feature distribution.
>
> For the challenges in the introduction:
> *  For the collaboration heterogeneity, the CSA module eliminates the agent feature heterogeneity with type-dependent projection and a learnable sampling mechanism that dynamically allocates the attention weights based on the feature distribution. The above operations enable the CSA module to distinguish the learning process of diverse agents to alleviate collaboration heterogeneity.
> *  For the feature misalignment, the CSA module enhances the target queries' features with local context and long-distance visual cues through an adaptive sampling method, improving the robustness against feature misalignment.
> ***
> **Q2**: Ablation of adaptive late fusion (ALF).
>
> **A2**: Insightful comments! As shown in lines 315-316, we replace the ALF module in Tab. 2 of the Main Paper with the pixel-wise addition strategy to perform the ablation study. The exclusive and common features are spatially complementary due to the previous decoupling operation. Therefore, the pixel-wise addition operation is able to partially combine the advantages of both features and present fine detection performance. The reason is that pixel-wise addition achieves the partial contribution of the ALF module since ALF is essentially a weighted version of the addition operation, as stated in lines 200-206. In this case, our ALF module adaptively integrates the perceptual contributions of the two features at each spatial location by adjusting the pixel-wise weights and, therefore, can further enhance the perception performance compared to the heuristic addition operation.
>
> Notably, we compare the other two fusion strategies in Tab. 1 of the Supplementary. The declining performance demonstrates the significant performance improvement of the proposed ALF module and its advantages in integrating perception contributions of different features.
> ***
> **Q3**:  About the dots in Fig. 8(b)&(c).
>
> **A3**:  The darker the colour of the forestgreen dots, the higher the attention weights of the corresponding sampled keypoints. From Fig. 8, the proposed exclusive and common spatial attention modules sample keypoints around the target query to extract object-related meaningful perception cues and assign larger attention weights to these keypoints. Moreover, they also reasonably sample long-range keypoints to extend the perception range of the ego agent. We will refine this description in the revision.
> ***
> **Q4**: About temporal dynamics in STCFormer.
>
> **A4**: As stated in lines 169-171, the ability of STCFormer to integrate temporal dynamics derives from our proposed Temporal Cross-Attention (TCA) module. From Fig. 3(a)&(b), the TCA module captures the temporal context from the ego agent's historical features and collaborator features to obtain the enhanced representation $H_{i}^{t}$, which is then passed to the subsequent Decoupled Spatial Attention component. Overall, STCFormer integrates both spatial semantic and temporal information, enabling more accurate perception.
> ***
> **Q5**: About limitations.
>
> **A5**: Constructive comments! We want to clarify that the assumption that features from different agents' LiDAR sensors are available follows previous works and consistent dataset protocols for a fair comparison. We agree that exploring the integration across sensor modalities with different characteristics via STCFormer is an essential research direction. However, the accessible dataset formats cause unavoidable limitations. For instance, current dataloaders do not support heterogeneous sensor data loading, and the labels required for image- and LiDAR-based detection paradigms are mismatched. We have done the best we can with the current scope. Moreover, we have further discussed the broad impacts and realistic limitations of How2comm in Sec. 5 and 6 of Supplementary, while indicating future directions for optimizing our system.

---

> > ### Comment · Reviewer_ToHv · 2023-08-18
> > **Official Comment by Reviewer ToHv**
> >
> > I appreciate the authors' efforts in the rebuttal. My concerns are somewhat addressed. I hope the authors would make sure to add clarification of the definition and limitations in the revised manuscript. I will stay with my original score.
> >
> > For Q5 in the rebuttal, I personally think the dataset is not an issue in this case since there are many existing multi-modal datasets for collaborative perception, such as the ones used in the paper (DAIR-V2X, etc). Meanwhile, One has to implement a new dataloader that can support heterogeneous sensor data loading can not be an excuse in academic research.

---

> > > ### Author Response · Authors · 2023-08-19
> > > **Official Comment by Authors to Reviewer ToHv**
> > >
> > > Dear Reviewer ToHv:
> > >
> > > Thanks for the response and the recognition of our rebuttal. We promise to add the definition of collaboration heterogeneity and the limitations of How2comm in the revision.
> > >
> > > We agree with your points. In future work, we will overcome the issues of multimodal data loading and label mismatch and further optimize How2comm to cope with collaboration heterogeneity caused by sensor discrepancies.
> > >
> > > Best regards,
> > >
> > > Authors

---

> ### Comment · Area_Chair_8hWx · 2023-08-18
>
> Dear Reviewer ToHv,
>
> Please read the author's rebuttal and other reviews and indicate whether your comments have been addressed. Thank you.
>
> Best, AC

---

### Official Review · Reviewer_aK4W · 2023-07-05

**Soundness:** 3 good
**Presentation:** 3 good
**Contribution:** 2 fair
**Rating:** 4
**Confidence:** 5

**Summary:**

This paper proposes How2comm, a collaborative perception framework, which seeks a trade-off between perception performance and communication bandwidth. This framework leverages a mutual information-aware communication to select the most informative messages to transmit and save bandwidth, where ego agents compute and send spatial and channel queries to neighbor agents. And a flow-guided delay compensation is applied to reduce temporal alignment. This paper designs Spatio-Temporal Collaboration Transformer to generate comprehensive features from spatial and temporal information. Experiments are conducted on three datasets Dair-V2X, V2XSet and OPV2V to validate the proposed methods.

**Strengths:**

1. This work considers the key problems in collaborative perception, including communication cost, transmission delay, localization error, designs a proper pipeline of collaboration and information processing  and get superior experiment results.
2. The logic and structure of the paper is clear.
3. The review and analysis of the field of study is comprehensive.

**Weaknesses:**

1. Many technical contributions claimed in this paper has been proposed in previous works, including requesting complementary information from neighbor agents to decrease communication costs [1], compensating latency with historical information [2] and fusing features with transformer [3]. This work appears to involve straightforward modifications of previous techniques and a system ensemble. So what is the essential technical novelty of the proposed methods? The authors need to clarify this point.

2. The main experiment results in Table.1 are significantly lower and inconsistent with previous work[4,5], which use the same Pointpillars backbone. The results seem not convincing.

3. The proposed framework includes more networks (transformer-based network), computations (importance map), and data (historical frames) than general collaborative perception methods, which may increase training costs and inference time. However, the performance seems not significant. There is a lack of evaluations of training cost, inference time and a trade-off between them and perception performance.

4. OPV2V and V2XSet are very similar datasets, the experiments should be conducted on datasets with different characteristics, like the V2X dataset V2X-Sim[5], the real-world V2V dataset V2V4real[6].

5. It lacks comprehensive comparison with previous methods that handle transmission delay. Previous work, like SyncNet [2] and V2X-ViT [3], has proposed to compensate for the latency with historical information. What are the advantages of the proposed flow-guided delay compensation?

[1] Hu Y, Fang S, Lei Z, et al. Where2comm: Communication-efficient collaborative perception via spatial confidence maps, Advances in neural information processing systems 2022.
[2] Lei Z, Ren S, Hu Y, et al. Latency-aware collaborative perception, European Conference on Computer Vision 2022.
[3] Xu R, Xiang H, Tu Z, Xia X, Yang M-H, Ma J, V2X-ViT: Vehicle-to-Everything Cooperative Perception with Vision Transformer, European Conference on Computer Vision 2022.
[4] Liu Y C, Tian J, Glaser N, et al. When2com: Multi-agent perception via communication graph grouping, Proceedings of the IEEE/CVF Conference on computer vision and pattern recognition. 2020.
[5]  Li Y, Ma D, An Z, et al. V2X-Sim: Multi-agent collaborative perception dataset and benchmark for autonomous driving, IEEE Robotics and Automation Letters, 2022.
[6] Xu R, Xia X, Li J, et al. V2v4real: A real-world large-scale dataset for vehicle-to-vehicle cooperative perception, Proceedings of the IEEE/CVF Conference on Computer Vision and Pattern Recognition. 2023.

**Questions:**

1. What is the meaning or definition of Collaboration Heterogeneity mentioned in section 1? What are the difficulties of fusing the information from different agents?
2. In Spatial-channel Message Filtering of section3.3, which dimension is the pooling functions applied to? The authors need to clarify it as well as all pooling functions in this paper.
3. Since the goal is to improve the perception performance of the ego agent with as little communication cost as possible, why maximizing the mutual information between the compressed and the vanilla feature  is necessary? If some information is useless to ego agent, it is unnecessary to keep them.
4. Does the communication cost include the queries and the indices of non-zero pixels? The authors need to clarify it to ensure a fair comparison.

**Limitations:**

The paper integrates a good collaboration perception system by overlaying various more or less known techniques. The core technical novelty is not clear. The experiments are also insufficient.

---

> ### Author Rebuttal · Authors · 2023-08-10
>
> Insightful comments! Below are some specific responses. The common questions can be found in the global response.
>
> **Q1**: About technical novelties.
>
> **A1**: Our three contributions fundamentally differ from previous works regarding design philosophy and structure modeling. We have highlighted contributions in lines 53-69 of Main Paper and Sec.1 of Supplementary.
> 1. Mutual information-aware communication (MIC)
> * Philosophy: Where2comm focuses on spatial sparsification without considering the redundant information in the channel dimension. In contrast, our MIC considers spatial-channel sparsification and the preservation of local critical semantics.
> * Structure: Where2comm is built on spatial confidence maps to perform spatial sparse communication. Our MIC performs space-channel sparse communication via customized attention queries. Also, MIC introduces mutual information maximization supervision to mitigate the loss of locally valuable semantics, which has been neglected in previous works.
> 2.  Flow-guided delay compensation (FDC)
> * Philosophy: SyncNet aims to predict future features with the received historical features, which may lead to large prediction errors since compressed features cause excessive information loss. In contrast, our FDC directly utilizes the complete local features within the collaborators to capture feature-level motion patterns.
> * Structure: SyncNet uses multi-scale LSTM structures to capture temporal cues for predicting the whole feature. Our FDC utilizes a simple flow generator to output feature flows and scale matrix, which can speed up convergence and achieves more accurate predictions at various delay levels.
> 3.  STCFormer
> * Philosophy: V2X-ViT exploits the vision transformer to aggregate collaborator features, ignoring contextual cues. Our STCFormer uses a unified structure to integrate spatio-temporal information and achieves more holistic feature fusion with decoupled design.
> * Structure: V2X-ViT uses per-location attention to learn pixel-wise attention weights among features. In our STCFormer, we design the novel temporal cross-attention to capture historical information, which V2X-ViT ignores. Further, we propose the importance-aware query initialization, keypoint sampling mechanism, and adaptive late fusion to achieve more effective collaboration.
> ***
> **Q2**: About main results.
>
> **A2**: We want to clarify that the inconsistent results are since our experimental setting has significant differences in bandwidth limitations from previous works. As stated in lines 241-242 of Main Paper and 245-250 of Supplementary, all experiments are constrained to $\approx$ 1 MB bandwidth consumption to reflect the narrow communication channels in more realistic and challenging V2V/X scenarios.
>
> In this case, we reproduce the baselines in Tab. 1 under bandwidth-limited noise settings based on the public codebase. Compared to the original reports, the decreased results of previous works prove that restricted communication resources can severely impact collaboration performance. Notably, How2comm outperforms existing models and improves the SOTA performance of AP\@0.7 by 7.4% on the real-world dataset DAIR-V2X.
>
> We will open the complete codebase for community progress if the paper is accepted.
> ***
> **Q3**: About transmission delay.
>
> **A3**: SyncNet utilizes received historical features to predict future features, which greatly consumes storage resources within the ego agent. Also, extracting the temporal semantics from the compressed features is difficult since the transmitted features have lost too much rich information. V2X-ViT uses delay-aware positional encoding to capture temporal information. This implicit compensation strategy would cause excessive accumulated errors under severe delay due to not utilizing historical features.
>
> In contrast,  our FDC strategy generates feature flows and the scale matrix with prediction ability at the collaborators' side and sends them to the ego agent. This strategy captures rich temporal information using complete local features within collaborators for more accurate prediction without consuming the ego agent's computation resources. From Fig. 6 in Main Paper, How2comm achieves SOTA performance at all delay levels of three datasets and maintains precision under severe delay, proving the FDC's advantages over existing methods.
> ***
> **Q4**: About pooling functions.
>
> **A4**:
> *  In line 122 of Sec. 3.3, average and max poolings in the spatial query are applied to the channel dimension. In line 125 of Sec. 3.3, average and max poolings are applied to the spatial dimension in the channel query. We provide more details in Fig.1 of Supplementary.
> *   In line 175, global average pooling is applied to the spatial dimension to compress the feature map.
> *   In line 181, max pooling is applied to the channel dimension to generate two-dimensional importance maps.
>
> We promise to add these minor explanations to the revision.
> ***
> **Q5**: About mutual information.
>
> **A5**: From lines 120-132, after the ego agent broadcasts request queries, collaborators eliminate unnecessary information for the ego agent via the spatial-channel message filtering matrix. In this case, the goal of mutual information supervision is not to maintain global information between vanilla features and compressed features, but to maximally preserve local critical semantics in the selected regions after message filtering, which avoids the potential loss of locally valuable information.
> ***
> **Q6**: About communication cost.
>
> **A6**: The ego agent broadcasts two request queries with sizes $(H \times W \times 1)$ and $(1 \times 1\times C)$. Since queries and non-zero-pixel indices are tiny compared to the communication volume of transmitted features, we do not consider their costs following Where2comm. Where2comm similarly ignores the overhead caused by the request map with size $(H \times W \times 1)$ and non-zero-pixel indices of transmitted features.

---

> > ### Author Response · Authors · 2023-08-17
> > **Official Comment by Authors**
> >
> > Dear Reviewer aK4W:
> >
> > We would like to thank the reviewer for taking the time to review our paper and for the comments.
> >
> > Please kindly let us know if anything is unclear. We truly appreciate this opportunity to clarify our work and shall be most grateful for any feedback you could give to us.
> >
> > Best regards,
> >
> > Authors

---

> ### Author Response · Authors · 2023-08-17
> **Official Comment by Authors to Reviewer aK4W**
>
> Dear Reviewer aK4W:
>
> As the discussion period is closing, we sincerely look forward to your feedback. We deeply appreciate your valuable time and efforts. It would be very much appreciated if you could once again help review our responses and let us know if these address or partially address your concerns and if our explanations are heading in the right direction.
> Please also let us know if there are further questions or comments about this paper. We strive to improve the paper consistently, and it is our pleasure to have your feedback!
>
> Best regards,
>
> Authors

---

> ### Comment · Area_Chair_8hWx · 2023-08-18
>
> Dear Reviewer aK4W,
>
> Please read the author's rebuttal and other reviews and indicate whether your comments have been addressed. Thank you.
>
> Best, AC

---

> > ### Comment · Reviewer_aK4W · 2023-08-18
> >
> > I deeply appreciate the authors' dedication to their responses and the thoroughness of their experiments. I am convinced that the proposed system is a powerful and comprehensive solution for multi-agent perception, a view that is also shared by the other reviewers.
> >
> > However, I must admit that I am not enthusiastic about accepting this work. For me, this work is more like an industrial product integrated on the assembly line. The contribution is mainly to skillfully integrate three techniques: mutual information-aware communication, flow-guided delay compensation and spatio-temporal collaboration transformer. For each of these three, it is obtained by adapting a widely recognized method with some necessary modifications, and the corresponding motivation, analysis, and new insights are very limited.
> >
> > Furthermore, I am concerned that this kind of all-inclusive works. This paper simultaneously considers three significant existing problems: communication redundancy, transmission delay and collaboration heterogeneity. Should each subsequent work also handle 3 or even more problems in a 8/9-page conference paper? Is this really good for a research field?
> >
> > Overall, I am fine if this is a journal paper, where the venue emphasizes system-level performances and the page restrictions are lenient . For NeurIPS, I expect to see a more compact presentation that focuses on innovative and enlightening methods.

---

> > > ### Author Response · Authors · 2023-08-19
> > > **Official Comment by Authors to Reviewer aK4W**
> > >
> > > Dear Reviewer aK4W:
> > >
> > > Thanks for your in-depth recognition of our rebuttal and thorough experiments. According to your earlier comments and current feedback, our rebuttal seems to have addressed most of your concerns. For the additional comments, we would like to clarify and discuss the following points.
> > >
> > > * We find it unconvincing and ambiguous to identify a work as "an industrial product integrated on the assembly line" without clear justification and explanation. The current collaborative perception system as a multi-stage practical application consists of multiple components/parts with different duties, which have been clarified and defined in previous works. Corresponding motivations and analyses for our customized contributions in How2comm have been highlighted several times in the Main Paper, Supplementary Material, and Rebuttal, which pragmatically address the weaknesses and shortcomings of existing methods. In this case, we note that you still argue that these insights are very limited but without further explanation or evidence. For each of these contributions, you claim they are modifications of widely recognized methods without specific references or explanations. In contrast, the authors in Rebuttal's A1 have clearly explained that our contributions differ entirely from the previous methods regarding design philosophy and structure modeling.
> > >
> > > * We think using interrogative sentences in the discussion phase to throw questions at the authors instead of clearly explaining your concerns is inappropriate and vague. As you know, there are multiple pressing challenges in the collaborative perception task. We sincerely appreciate that some previous efforts have focused on addressing one of these and have made promising progress. However, simultaneous consideration (but not mandatory) of several possible optimization directions for this realistic task subject to multiple interfering factors can improve the system's robustness and breakthrough performance bottlenecks. One fact is that existing prominent works are derived from conference papers. They have begun considering addressing multiple challenges simultaneously and providing insightful contributions to community development. As an intuitive example, the pioneer V2X-ViT [1] addressed agent heterogeneity challenge through the heterogeneous multi-agent self-attention component, feature mismatch challenge through the multi-scale window attention component, and temporal asynchrony challenge due to transmission delay through the delay-aware positional encoding component. Objectively, it is their choice whether subsequent works address multiple challenges. The page count of a paper is not an excuse to limit the potential work.
> > >
> > > * In the comments, the other three reviewers highly endorsed this work's clear presentation, coherent writing, good organization, and novel approach. In addition, we have shown further details of the system pipeline and the design of the main modules in Sections 2 and 3 of the Supplementary Material, respectively. We argue that there is no direct correlation between the evaluation of a work and whether it belongs to a conference or a journal paper.
> > > Overall, we would like to emphasize that 1) we sincerely appreciate your valuable time and efforts, and 2) we respect your subjective viewpoints; however, we must clarify our work to avoid misleading and confusing other readers and reviewers.
> > >
> > > Best regards,
> > >
> > > Authors
> > >
> > > [1] Xu R, Xiang H, Tu Z, Xia X, Yang M-H, Ma J, V2X-ViT: Vehicle-to-Everything Cooperative Perception with Vision Transformer, European Conference on Computer Vision 2022.

---

### Official Review · Reviewer_fcX9 · 2023-07-06

**Soundness:** 3 good
**Presentation:** 3 good
**Contribution:** 3 good
**Rating:** 6
**Confidence:** 4

**Summary:**

The paper introduces three key contributions in tackling the aforementioned challenges. Firstly, the authors devise a mutual information-aware communication mechanism that maximizes the sharing of informative features among collaborators. They employ spatial-channel filtering for efficient feature sparsification and effective communication. Secondly, a flow-guided delay compensation strategy is presented to predict future characteristics from collaborators, addressing feature misalignment caused by temporal asynchrony. Lastly, a pragmatic collaboration transformer is introduced to integrate spatial semantics and temporal context clues, enhancing the overall perception capabilities among agents. The proposed framework, How2comm, is thoroughly evaluated on three LiDAR-based collaborative detection datasets in both real-world and simulated scenarios. The experiments conducted are comprehensive and demonstrate the superiority of How2comm over existing methods. The effectiveness of all vital components within the framework is also validated, providing solid evidence for the contributions made.

**Strengths:**

1. Novel Approach: How2comm addresses the challenges faced in multi-agent collaborative perception by introducing innovative techniques, such as mutual information-aware communication, flow-guided delay compensation, and pragmatic collaboration transformer.

2. Comprehensive Evaluation: The framework is rigorously evaluated on various datasets, including real-world and simulated scenarios, establishing its superiority over existing methods.

3. Clear Presentation: The paper effectively explains the proposed framework and its components, making it accessible to readers with varying levels of expertise.

**Weaknesses:**

Here are some suggestions:

1. It would be beneficial for the authors to discuss any limitations or potential areas for improvement in the How2comm framework, providing avenues for future research.

2. The authors are suggested to add some missing literature to be more comprehensive.

[1] Li, Y., Ma, D., An, Z., Wang, Z., Zhong, Y., Chen, S. and Feng, C., 2022. V2X-Sim: Multi-agent collaborative perception dataset and benchmark for autonomous driving. IEEE Robotics and Automation Letters, 7(4), pp.10914-10921.

[2] Yu, H., Luo, Y., Shu, M., Huo, Y., Yang, Z., Shi, Y., Guo, Z., Li, H., Hu, X., Yuan, J. and Nie, Z., 2022. Dair-v2x: A large-scale dataset for vehicle-infrastructure cooperative 3d object detection. In Proceedings of the IEEE/CVF Conference on Computer Vision and Pattern Recognition (pp. 21361-21370).

[3] Xu, R., Xia, X., Li, J., Li, H., Zhang, S., Tu, Z., Meng, Z., Xiang, H., Dong, X., Song, R. and Yu, H., 2023. V2v4real: A real-world large-scale dataset for vehicle-to-vehicle cooperative perception. In Proceedings of the IEEE/CVF Conference on Computer Vision and Pattern Recognition (pp. 13712-13722).

[4] Yu, H., Yang, W., Ruan, H., Yang, Z., Tang, Y., Gao, X., Hao, X., Shi, Y., Pan, Y., Sun, N. and Song, J., 2023. V2X-Seq: A Large-Scale Sequential Dataset for Vehicle-Infrastructure Cooperative Perception and Forecasting. In Proceedings of the IEEE/CVF Conference on Computer Vision and Pattern Recognition (pp. 5486-5495).

[5] Li, Y., Zhang, J., Ma, D., Wang, Y. and Feng, C., 2023, March. Multi-robot scene completion: Towards task-agnostic collaborative perception. In Conference on Robot Learning (pp. 2062-2072). PMLR.

[6] Su, S., Li, Y., He, S., Han, S., Feng, C., Ding, C. and Miao, F., 2023. Uncertainty quantification of collaborative detection for self-driving. ICRA.



**Questions:**

The authors are suggested to provide a video demonstration of the results.

**Limitations:**

N.A.

---

> ### Author Rebuttal · Authors · 2023-08-04
>
> We thank the reviewer for recognizing our contribution of novel approach, comprehensive evaluation, and clear presentation. We present our responses below.
>
> **Q1**: Discussion of  any limitations or potential areas for improvement.
>
> **A1**: 1. Thanks for your constructive comments! We have discussed the limitations and future work of the proposed framework in lines 340-342 of the main manuscript. Concretely,
> *  The current work only exploits the short-term historical frames. In future work, we plan to expand the utilization of temporal information to long-term point cloud sequences.
> *  We will explore optimizing the feature flow prediction with uncertainty estimation.
> 2. Moreover, we have discussed the **Broader Impacts** of our How2comm in Section 5 (lines 356-367) of the **Supplementary Material**. Concretely,
> *  The positive impact of this work is that How2comm can leverage vehicular communication
> technology to facilitate information sharing and enhance the vehicles' perception ability of the
> surrounding environment, reducing the incidence of traffic accidents.
>  *  The collaborative perception ability provided by How2comm can be used to monitor real-time traffic conditions and improve traffic efficiency.
> *  The illegal personnel may invade the vehicle communication channel to attack the collaborative perception system, leading to the vehicles' misjudgment of the surrounding environment or malicious remote control.
> *  How2comm facilitates Vehicle-to-Vehicle/Everything (V2V/X)-oriented multi-agent communication applications in autonomous driving.
> 3. Further, we have discussed other **Realistic Limitations** and indicated future research directions to optimize our system in Section 6 (lines 368-387) of the **Supplementary Material**. Concretely,
> *  For the attack issue, How2comm could improve the robustness against potential adversarial attacks by introducing an adversarial training paradigm. In addition, by focusing on decoupled spatial regions and discovering perceptually critical information, How2comm is relatively less likely to be attacked.
> *  For the data bias issue, How2comm has demonstrated its preliminary generalization on real-world and simulated datasets with different configurations and sensor facilities. In future work, How2comm could disentangle causal relationships among variables in the collaborative perception pipeline with the assistance of causal inference techniques, further enhancing its ability to generalize to more complex real-world scenarios.
> *  For the domain gap issue, How2comm could hopefully achieve sim2real adaptation and inter-agent adaptation by introducing an unsupervised sim2real domain bridging mechanism.
> ***
> **Q2**: Addition of missing literature.
>
> **A2**: Many thanks for the insightful suggestions! We promise to add all the literature you listed in the revision for a more comprehensive overview and presentation.
> ***
> **Q3**: Provision of a video demonstration of the results.
>
> **A3**:  Valuable proposal! In our submitted **Supplementary Material**, we have provided a video demonstration of an intuitive performance comparison between How2comm and the previous SOTA models. You can freely browse the *video.mp4* file. The video demonstration provides the following observations.
>
> How2comm has fewer regression misalignments and fewer undetected objects, resulting in a more robust performance. In comparison, the previous methods fail to tackle fast-moving objects and perceive remotely occluded regions. The indisputable evidence is that some ground truth bounding boxes are not well matched with their detection results.

---

> > ### Comment · Reviewer_fcX9 · 2023-08-22
> >
> > Thank authors for the response and I will keep my original rating.

---

> ### Comment · Area_Chair_8hWx · 2023-08-18
>
> Dear Reviewer fcX9,
>
> Please read the author's rebuttal and other reviews and indicate whether your comments have been addressed. Thank you.
>
> Best, AC

---

### Author Rebuttal · Authors · 2023-08-10

We thank all reviewers for their time and effort. Here we provide detailed responses to common questions.

**Q1**: Trade-off between training cost, inference time, and perception performance.

**A1**:
* We compare the evaluated models regarding space-time complexity, inference time, and detection performance across the five datasets in Tab. 1 of the **response pdf**. Results of the OPV2V Culver City and V2V4Real datasets are presented in Tab. 2 in Supplementary and Tab. 3 of the response pdf. We adopt three mainstream metrics: Parameter number, FLOPs, and Fps, and use the same hyperparameter setting as the Main Paper. Intuitively, How2comm outperforms previous SOTAs on five datasets. In particular, How2comm improves the SOTA performance of AP\@0.7 by 7.4% and 6.5% on DAIR-V2X and V2V4Real, respectively, showing significant improvements.
* Concretely, How2comm outperforms the complex transformer-based V2X-ViT in terms of Parameter number, FLOPs, and Fps on five datasets. Moreover, How2comm reduces the Parameter number and FLOPs compared to CoBEVT and improves the Fps on most datasets compared to Where2comm. The above results indicate that How2comm is competitive and comparable in space-time complexity and inference time compared to the SOTAs. Overall, the above results prove that How2comm can achieve a better tradeoff among training cost, inference time, and perception performance than most SOTA methods.
* The reasonable explanations are:
1. From lines 189-190, we introduce importance-aware query initialization in STCFormer to guide decoupled spatial attention modules to focus on the potential foreground objects. This approach uses the sparsity of point cloud data to reduce query number and computing complexity while maintaining perception performance.
2. From lines 179-181, the importance maps are obtained by simple sigmoid activation, max pooling, and $f_{gen}$ implemented by 1*1 convolution. The complexity introduced by these operations is negligible.
***
**Q2**:  Discussion of datasets.

**A2**: We want to clarify that OPV2V and V2XSet datasets differ greatly regarding agent type, agent count, and scenario characteristics, which V2X-ViT confirms. Also, Tab. 2 of the Supplementary indicates the excellent performance of How2comm on the OPV2V Culver City validation set, which provides challenging driving scenarios that are close to the real world.

Further, we provide the results on the V2V4Real dataset in Tab. 3 of the **response pdf** for a more holistic comparison. We employ the same delay settings and bandwidth constraints as the Main Paper, and localization noises are not added since this dataset has natural GPS errors. Intuitively, How2comm outperforms existing SOTA models and improves the SOTA performance of AP\@0.7 by 6.5%. Overall, How2comm improves SOTA performance on all five datasets, showing our method's superiority in real-world and simulated scenarios. Limited rebuttal time did not allow us to conduct experiments on the V2X-Sim; we will add comparative results in the revision.
***
**Q3**: Definition of collaboration heterogeneity and difficulties for fusing information from different agents.

**A3**: Current heterogeneity in collaborative perception systems may come from two aspects: heterogeneous sensor modalities and the same type of sensor's configuration discrepancies. We consider the latter due to some practical constraints. In this paper, collaboration heterogeneity means the heterogeneity in feature-level collaboration due to LiDAR configuration discrepancies from distinct agents (e.g., vehicles and infrastructures), including different LiDAR densities, distributions, reflectivities, and noise interference.

In this case, the following difficulties exist for feature fusion in the collaboration process:
* How to integrate object-related representations from complementary regions of heterogeneous agents to extend the perception range.
* How to mitigate the feature map gap due to sensor configuration differences in the common perception regions.
* How to overcome feature map misalignment caused by collaborative noises. e.g., localization errors.

Motivated by the above difficulties, we propose the novel STCFormer to tackle the existing challenges.
***
**Q4**: Discussion of ablation studies.

**A4**: The ablation results on key hyperparameters and structure choices are presented in Tab. 1 of the Supplementary. Moreover, we provide more ablation results for decoupled designs and complete modules in Tab. 2 of the **response pdf**.

First, we set both $\mathcal{E,C}$ to 1 and use the same attention module in both branches. Intuitively, Full Model achieves better detection performance on all five datasets than the version without Decoupled Design, proving that How2comm effectively integrates the perception advantages from exclusive and common regions via two decoupled branches and yields more robust visual representations against collaboration heterogeneity. Without explicitly defining exclusive and common features, spatial attention will indiscriminately sample the entire region due to the lack of guidance from prior information, which may introduce excessive collaborator noises and impede bridging to collaboration heterogeneity.

Meanwhile, we show the ablation results for the complete modules. We directly remove the FDC module and replace MIC and STCFormer with a compression-based communication method and a 1*1 convolution-based fusion, respectively. The degraded performance provides the following observations:

1. Our MIC module can facilitate detection performance by retaining more critical semantic information under limited bandwidth.
2. When FDC is removed, the feature misalignment caused by temporal asynchrony significantly impact the feature fusion process and harm the precision.
3. STCFormer enables more valuable spatio-temporal semantics to be extracted from collaborator-shared features and historical frames than 1*1 convolution-based fusion.

---

### Decision · Program_Chairs · 2023-09-21

**Decision:**

Accept (poster)

**Comment:**

The paper proposes an efficient cooperative perception method with robust modules designed to accommodate a variety of real-world factors. Reviewers fcX9, ToHv, and kdrk are enthusiastic about the concept, affirming that the paper is well-written, sufficiently novel, and addresses potential real-world challenges effectively. Reviewer aK4W expressed reservations about the novelty and experimental sections of the paper. While the ACs partially agree with reviewer aK4W's concerns about similarities between certain portions of the paper and previous works, they appreciate the innovative elements within the paper. After reading the comments and the draft, the ACs recommend to accept this paper.